# Perceptually Constrained Precipitation Nowcasting Model

**Wenzhi Feng** [1 2 3]  **Xutao Li** [1 2 3]  **Zhe Wu** [2]  **Kenghong Lin** [1 3]  **Demin Yu** [1 3]  **Yunming Ye** [1 2 3]  **Yaowei Wang** [1 2]

## Abstract

Most current precipitation nowcasting methods aim to capture the underlying spatiotemporal dynamics of precipitation systems by minimizing the mean square error (MSE). However, these methods often neglect effective constraints on the data distribution, leading to unsatisfactory prediction accuracy and image quality, especially for long forecast sequences. To address this limitation, we propose a precipitation nowcasting model incorporating perceptual constraints. This model reformulates precipitation nowcasting as a posterior MSE problem under such constraints. Specifically, we first obtain the posteriori mean sequences of precipitation forecasts using a precipitation estimator. Subsequently, we construct the transmission between distributions using rectified flow. To enhance the focus on distant frames, we design a frame sampling strategy that gradually increases the corresponding weights. We theoretically demonstrate the reliability of our solution, and experimental results on two publicly available radar datasets demonstrate that our model is effective and outperforms current state-of-the-art models.

## 1. Introduction

Weather forecasting is an important spatial and temporal prediction task that affects the comfort and safety of people's daily lives and profoundly affects a number of key industries, such as agriculture, aviation, navigation, and energy supply. Accurate weather forecast plays a crucial role in safeguarding people's lives and promoting economic and social development.

[1]School of Computer Science and Technology, Harbin Institute of Technology, Shenzhen, China [2]Pengcheng Laboratory, Shenzhen, China [3]Shenzhen Key Laboratory of Internet Information Collaboration, School of Computer Science and Technology, Harbin Institute of Technology, Shenzhen, China. Correspondence to: Xutao Li <lixutao@hit.edu.cn>.

*Proceedings of the 42^{nd} International Conference on Machine Learning*, Vancouver, Canada. PMLR 267, 2025. Copyright 2025 by the author(s).

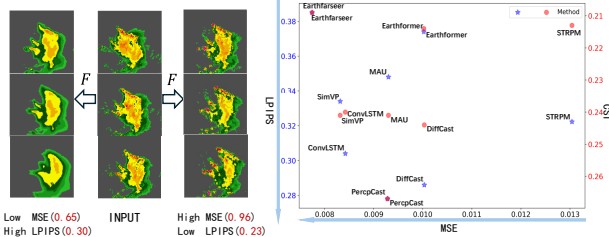

*Figure 1.* Illustration of the accuracy-perception tradeoff in precipitation prediction. Historical data (center) are input into two state-of-the-art algorithms for forecasting precipitation. The scatter plot demonstrates the performance of current state-of-the-art methods in terms of the accuracy-perception tradeoff, with CSI as the primary metric for precipitation prediction.

Precipitation nowcasting aims to predict rainfall within the next six hours, characterized by high spatiotemporal resolution and uncertainty about the future. Most of the current precipitation prediction methods are dedicated to improving the accuracy where the forecast image quality is neglected. However, image quality can reflect to a certain extent whether the model can simulate the real-world physical state, which is especially important in long-term prediction. As shown in the left part of Figure 1, the model-smoothed prediction is obtained on the left side, and the physically realistic prediction is obtained on the right side.

Existing precipitation nowcasting models can be broadly divided into two categories, deterministic models(Bai et al., 2022; Ning et al., 2023; Wu et al., 2021) and probabilistic models(Chang et al., 2022; Luo et al., 2022; Zhang et al., 2023). Deterministic models such as ConvLSTM(Shi et al., 2015), Transformer(Vaswani et al., 2017), SimVP(Gao et al., 2022b), are designed to predict future sequences by generalizing spatiotemporal patterns from historical sequences. These models reduce the MSE and tend to produce an average prediction over all possible future states in the case of uncertainty. However, this line of methods cannot match the expected distribution of ground-truth data and thus suffers from the problems of absent details and intensity decay, which is often shown as a better accuracy performance (lower MSE) but a worse perceptual performance (higher

LPIPS).On the other hand, probabilistic models, such as STRPM(Chang et al., 2022) and PreDiff(Gao et al., 2024), predict future image sequences by learning their distribution from current observations. While these methods can produce predictions with better detail and intensity (lower LPIPS), they often struggle with accuracy in positioning (higher MSE), as the space of samples following the target distribution is infinite.

In long-sequence precipitation nowcasting, neither mean squared error nor perceived quality fully measures forecast accuracy, as shown on the right side of Figure 1. Accurate forecasts must satisfy both criteria, along with other key metrics (e.g., CSI). Therefore, achieving high-quality predictions with low error is a key challenge.

In this paper, we propose a novel perceptual constrained precipitation forecast model, named PercpCast, which effectively addresses the limitations of the existing methods by constraining the data distribution of the prediction sequence, thereby improving the accuracy of the resulting predictions. First, we use a precipitation estimator to obtain the approximate posteriori mean sequence of the predicted target. Then, we train a rectified flow model to predict the straight line between the corresponding posteriori mean sequence and the target sequence. Specifically, we use the autoregressive structure of ConvLSTM as a precipitation estimator to capture the underlying spatiotemporal dynamics of the precipitation system and obtain a continuous sequence of posteriori mean values. However, affected by the cumulative error, the distribution difference between the frames forming the posteriori mean sequence becomes more significant with time, so the path from each predicted frame to the target frame is different. To better model the transition from predicted frames to target frames, we design a frame-sampled rectified flow model that models each predicted frame individually and provides weighted scheduling that lets the model focus on frames with longer prediction times. The entire model is trained in an end-to-end manner, and the set of a posteriori mean sequences is fed into the flow model as an initial state to solve the ODE, which avoids falling into a local optimum.

The main contributions of this paper are summarized as follows:

- We propose an end-to-end precipitation prediction model based on perceptual constraints by constraining the data distribution while achieving better prediction error and image quality.

- We propose a novel video-rectified flow model that uses sampling training to simulate the path from the posteriori mean sequences to the real sequences and employs lpips loss for backward propagation to ensure perceptual consistency.

- We propose a temperature-distance weighted scheduling that lets the model focus on frames at the tail of the sequence.

- Our method achieves optimal performance on both Sevir and MeteoNet datasets.

## 2. Related Work

**Precipitation Nowcasting**: For precipitation nowcasting, deterministic models can effectively capture the overall trend of precipitation movement by modeling the posteriori mean sequence of future precipitation. The ConvLSTM(Shi et al., 2015) model integrates convolution operators into the LSTM and uses extrapolation loops to maintain consistency of predicted motion. Earthformer(Gao et al., 2022a) builds an encoder-decoder model with Transformer structures, which replaces the original attention with elaborate cube attention. SimVP(Gao et al., 2022b) uses a simplified encoder and decoder structure based on convolutional neural networks, significantly improving computational efficiency. MAU(Chang et al., 2021) improves the precipitation forecast by mining the current and historical spatial states to extend the time horizon. PhyDNet(Guen & Thome, 2020) uses partial differential equations (PDEs) to disentangle PDE dynamics from unknown complementary information and performs PDE-constrained prediction in the latent space. However, all these methods suffer from blurring and high echo fading issues. Probabilistic models make predictions by modeling the data distribution of future precipitation. DGMR(Ravuri et al., 2021) uses adversarial training to constrain prediction distribution as close to the real precipitation by introducing spatial and temporal discriminators. PreDiff(Gao et al., 2024) constructs a diffusion model of the potential space and designs a priori knowledge modules to adjust the predicted distributions. Diffcast(Yu et al., 2024) simulates local random variations by introducing a residual diffusion mechanism. This kind of model helps to generate realistic details in a predictive framework but performs poorly on predictive metrics.

**Image Perception Quality**: Image Perceptual Quality Assessment is a method of quantifying the quality of an image, which is widely used in image reconstruction, image compression, and image generation. In image enhancement, PCSGAN(He et al., 2024) develops a generative adversarial network with perceptual constraints, which imposes specific positional and structural constraints on the image and achieves a better sense of image realism. In image restoration, PiRN(Jaiswal et al., 2023) proposes physically integrated restoration networks that introduce physically based simulators and stochastic refiners to improve its perceptual quality. In the field of video generation, STRPM(Chang et al., 2022) introduces an additional perceptual loss in the

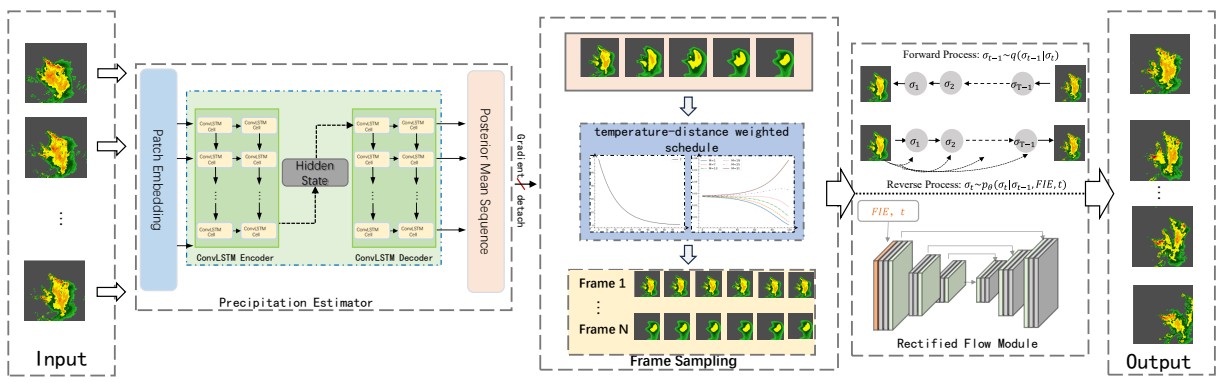

*Figure 2.* An overview of the proposed PercpCast for precipitation nowcasting. The model first obtains the posteriori mean sequences of future precipitation employing a precipitation estimator, then constrains the posteriori distribution to mean sequences using the rectified flow. It designs a frame-sampling strategy to increase the attention paid to the long-term prediction frames.

generative adversarial network, which is used to constrain the feature representations of the generated image in the discriminator, thereby improving the perceptual quality of the prediction. In the field of super-resolution, TDPN(Cai et al., 2022) defines a new hybrid loss to enhance the boundary information recovery, which helps to overcome the suboptimums of MAE loss.

## 3. PROBLEM STATEMENT

Precipitation nowcasting can be formulated as a spatiotemporal sequence prediction problem(Gao et al., 2022a; Shi et al., 2015; Tan et al., 2023), which is defined as

$$\min_{p_{\hat{Y}|X}} \mathbb{E}[\|Y - \hat{Y}\|^2]. \tag{1}$$

Specifically, given an initial precipitation sequence $X = [x_i]_{i=1}^{L_{in}}, x_i \in \mathbb{R}^{H \times W \times C_{in}}$, the model predicts that the future sequence $Y = [y_i]_{i=1}^{L_{out}}, y_i \in \mathbb{R}^{H \times W \times C_{out}}$, where $L_{in}$ and $L_{out}$ denote the lengths of the input and predicted frames, respectively. $H$ and $W$ denote the heights and widths of each frame. $C_{in}$ and $C_{out}$ denote the number of channels in the input and predicted frames at each time point. In addition, we use the subscript $i$ to denote the $i$-th frame, e.g. $y_i$.

The posteriori mean sequence of the model can be obtained from Eq.1. However, as the prediction time increases, the difference between the distribution of the predicted frame and the distribution of the target frame becomes larger and larger. To solve this problem, we introduce a perceptual constraint to align the predicted distribution with the target distribution. As a result, we reformulate and define the precipitation task as follows:

$$\min_{p_{\hat{Y}|X}} \mathbb{E}[\|Y - \hat{Y}\|^2] \quad \text{s.t.} \quad p_{\hat{Y}} = p_Y. \tag{2}$$

We assume that $Y$ and $\hat{Y}$ are independent of each other given $X$. In other words, the predicted values depend only on the historical observations, and there exists $\hat{Y}^* = \mathbb{E}[Y|X = x]$, which is the optimal or approximately optimal solution for the posteriori mean. Furthermore, inspired by (Freirich et al., 2021), we give the optimal transport form of Eq.3. as

$$\mathbb{E}[\|Y - \hat{Y}^*\|^2] + \mathbb{E}[\int_0^1 \|(Y - Z_0) - v_\theta(Z_t, t)\|^2 \, dt], \tag{3}$$

where $Z_0 = \hat{Y}^*, Z_t = tY + (1-t)Z_0$. The detailed analysis is given in Appendix A.

## 4. The Proposed Model

To address the accuracy and perception dilemma, we propose a perceptually constrained precipitation nowcasting model, as shown in Figure 2, which is composed of three main components: (1) a precipitation estimator to capture the spatial and temporal dynamics of the precipitation system and to obtain a sequence of future posteriori mean values; (2) a rectified flow network that aligns the distribution of the posteriori mean sequences to the target distribution, to simulate physical reality. (3) an end-to-end training scheme. Next, we elaborate on the proposed PercpCast method by introducing the three components, respectively.

### 4.1. Precipitation Estimator

To capture the evolving precipitation patterns from historical data and to obtain smoother, more continuous forecasts, we utilize ConvLSTM as the precipitation estimator.

Specifically, first we apply the patch embedding to the history sequence $X = [x_i]_{i=1}^{L_{in}} \in \mathbb{R}^{L_{in} \times H \times W \times C}$ to get the embedding $X_{patch} = [x_i]_{i=1}^{L_{in}} \in \mathbb{R}^{L_{in} \times H/4 \times W/4 \times 16C}$, and then $X_{patch}$ is fed into the image encoder to obtain $X_{enc} = [x_i]_{i=1}^{L_{in}} \in \mathbb{R}^{L_{in} \times H/4 \times W/4 \times 16C}$, next we feed $X_{enc}$ into the

ConvLSTM and decode it through the decoder to get the estimated value $\hat{Y}^* = [x_i]_{i=1}^{L_{in}} \in \mathbb{R}^{L_{in} \times H/4 \times W/4 \times 16C}$.

Finally, the estimated value is optimized using the Mean Square Error criterion:

$$\mathcal{L}_{pe} = \mathbb{E}\left[\|Y - \hat{Y}^*\|^2\right]. \tag{4}$$

Eq.4 illustrates that the estimated value $\hat{Y}^*$ represents the posteriori mean sequence of the future precipitation. Next, we detail the construction of perceptual constraints based on this posteriori mean sequence.

### 4.2. Frame-sampled Rectified Flow Model

In Section 3, we demonstrated that the precipitation forecasting problem under perceptual constraints can be solved by Eq.3. However, long-term spatiotemporal sequence prediction is highly susceptible to future uncertainty. This leads to variations in the distribution of each frame in the posterior mean sequence generated by the precipitation estimator over time. For instance, as $x \to x_0$, the distribution of $x$ becomes increasingly realistic as it approaches $x_i$, while as $x \to x_T$, where $T = L_{out}$, the gap between the distributions of $x$ and $x_i$ widens, resulting in smoother predictions. We attempted to transfer the distribution of the posterior mean sequence to the target sequence using a rectified flow model. However, since the posteriori mean sequence consists of a mixture of distributions, modeling the entire target sequence becomes challenging due to the influence of mode overlap. To more effectively model the transition from the posteriori mean sequence to the target sequence, we designed a rectified flow model enhanced with frame sampling.

Specifically, for the posteriori mean sequence obtained by the precipitation estimator, we first stop the gradient derivation to ensure that $\hat{Y}^* \to \hat{Y}$ follows a Markov process. Next, we reshape the posteriori mean sequence $\hat{Y}^* \in \mathbb{R}^{B \times T \times C \times H \times W}$ into $\hat{Y}^* \in \mathbb{R}^{BT \times C \times H \times W}$, then we construct the Frame Indexed Embedding (FIE) based on the order of the frames in the posteriori mean sequence. Finally, the FIE and the reshaped posteriori mean sequence $BT * C * H * W$ are concatenated and fed into the U-Net structure to learn the velocity field from the posteriori mean sequence to the target sequence at all frame indexes, with the objective function as follows:

$$\mathbb{E}\left[\int_0^1 \|(Y - Z_0) - v_\theta(Z_t, t, FIE)\|^2 \, dt\right]. \tag{5}$$

Where $Y \in \mathbb{R}^{BT \times C \times H \times W}$, $Z_0 \in \mathbb{R}^{BT \times C \times H \times W}$, $FIE \in \mathbb{R}^{BT \times d}$, $d$ is the embedding dimension.

Furthermore, to encourage the model to place greater emphasis on frames further away from $x_0$, we adopt a temperature-distance weighted frame sampling strategy. In this approach,

only one frame is sampled for each training iteration, with the probability of selection increasing with the distance from $x_0$, thereby assigning different training weights to the various prediction frames. Specifically, we first compute the base weights for each of the $T$ prediction frames:

$$w_i = \exp(k \cdot i) \tag{6}$$

$k$ is a fixed parameter that controls the rate of weight growth, $i \in [0, T-1]$ is the number of each of the frames, and then the weights are adjusted using a dynamic temperature $\tau$,

$$w_i' = w_i^{1/\tau} \tag{7}$$

where, depending on the training process, the dynamic temperature can be linear:

$$\tau(t) = \tau_{\text{start}} + (\tau_{\text{end}} - \tau_{\text{start}}) \cdot \frac{t}{t_{\max}} \tag{8}$$

or exponential:

$$\tau(t) = \tau_{\text{start}} \cdot \left(\frac{\tau_{\text{end}}}{\tau_{\text{start}}}\right)^{\frac{t}{t_{\max}}} \tag{9}$$

we then normalize the adjusted weights to a probability distribution $p_i$ and obtain the cumulative probability distribution (CDF),

$$p_i = \frac{w_i'}{\sum_{j=0}^{T-1} w_j'} \tag{10}$$

$$\text{CDF}_i = \sum_{j=0}^{i} p_j, \quad i \in [0, T-1]. \tag{11}$$

For sampling, a random number $r$ is generated from a uniform distribution $r \sim U(0, 1)$ and then the random number $r$ is mapped to the corresponding frame sequence number using a cumulative probability distribution function (CDF).

$$\text{sampled\_index(sid)} = \min\{i \mid \text{CDF}_i \geq r\} \tag{12}$$

The final rectified flow model objective function with frame sampling can be written as

$$\mathcal{L}_{rf} = \mathbb{E}\left[\int_0^1 \left\|Y^{\text{sid}} - Z_0^{\text{sid}} - v_\theta(Z_t^{\text{sid}}, t, FIE^{sid})\right\|^2 dt\right]. \tag{13}$$

Additionally, to ensure high-quality prediction results and prevent degradation of the generated model, we employ the Lpips loss to constrain the velocity field $v_\theta$, enabling the model to maintain strong perceptual consistency at each step of the reversal process.

$$\mathcal{L}_{\text{lpips}} = \sum_l \alpha_l \|\phi_l(Z_0^{\text{sid}} + v_\theta(Z_t^{\text{sid}}, t, FIE^{sid}) - \phi_l(Y^{\text{sid}})\|_2^2 \tag{14}$$

where $\phi_l(\hat{Y}^{\text{sid}})$ represent the image features extracted by the deep neural network (such as VGG or AlexNet) at the $l$-th layer, $\alpha_l$ is the weighting coefficient for each layer's feature.

**Algorithm 1** Training Process

> **Input:** data $X = [x_i]_{i=1}^{L_{in}}, x_i \in \mathbb{R}^{H \times W \times C}$
> **initial:** $k, \tau$
> **repeat**
>    $h = x_0$
>    **for** $i = 1$ **to** $L_{in} + L_{out}$ **do**
>      $h = PatchEmbeding(h)$
>      $\hat{y}^*_{-L_{in}+i} = ConvLSTM(h; \theta_1)$
>      $h = \hat{y}^*_{-L_{in}+i}$
>    **end for**
>    Compute loss $\mathcal{L}_{pe}$
>    Sampling a frame $\hat{Y}^{\text{sid}}$ form $[\hat{y}^*_1, \hat{y}^*_2...\hat{y}^*_{L_{out}}]$ by sampled_index
>    detach gradient of $\hat{Y}^{\text{sid}}$
>    Sampling flow step $t \sim \mathcal{U}[0, 1]$
>    Forward Process: $z_t = t(Y^{\text{sid}}) + (1 - t)\hat{Y}^{\text{sid}}$
>    Backward Process $v_t(z_t; \theta_2) = Y^{\text{sid}} - \hat{Y}^{\text{sid}}$
>    Compute loss $\mathcal{L}_{rf}$
>    Update $\tau$
>    Update Parameters $\theta_1, \theta_2$
> **until** Loss convergence
> **RectFlow**(optional): $(Z_0^0, Z_1^0) = (\hat{Y}^*, \hat{Y})$.
> **repeat**
>    **Reshape:** Merge Channel Batch and Channel Time
>    Sampling flow step $t \sim \mathcal{U}[0, 1]$
>    Forward Process: $z_t = t(Z_1^0) + (1 - t)Z_0^0$
>    Backward Process $v_t(z_t; \theta_2) = \hat{Y} - \hat{Y}^*$
>    Compute loss $\mathcal{L}_{rf}$
> **until** Loss convergence

**Algorithm 2** Inference Process

> **Input:** data $X = [x_i]_{i=1}^{L_{in}}, x_i \in \mathbb{R}^{H \times W \times C}$
> $h = x_0$
> **for** $i = 1$ **to** $L_{in} + L_{out}$ **do**
>    $h = PatchEmbeding(h)$
>    $\hat{y}^*_{-L_{in}+i} = ConvLSTM(h)$
>    $h = \hat{y}^*_{-L_{in}+i}$
> **end for**
> **Reshape:** Merge Channel Batch and Channel Time
> **if** RectFlow **then**
>    $\hat{z} = v_t(\hat{Y}^*, t)$
>    $z_0 = z_0 + \hat{z}$
> **else**
>    **while** t in 0-N **do**
>      $\hat{z} = v_t(z_0, t)$
>      $z_0 = z_0 + \frac{1}{N}\hat{z}$
>    **end while**
> **end if**
> $\hat{Y} = z_0$
> **Reshape:** Split Channel Batch and Channel Time
> **Return:** $\hat{Y}$

paths by reducing them through a vector field, and computes the loss, and finally chooses whether to rectify or not. In the inference phase, the process is similar. The precipitation estimator first generates the posterior mean sequence $\hat{Y}^*$, and then the time dimension of $\hat{Y}^*$ is converted to a batch dimension. This input is fed into the rectified flow model, where predicted values with perceptual constraints are obtained via vector field iteration. The final output is then produced by separating the time steps from the batch dimension.

## 5. Experiments

To validate the effectiveness of the proposed model, we conduct experiments on two real precipitation datasets and perform an ablation study to analyze the contributions of the individual components within the model. The experimental results lead to the following key conclusions: 1) The proposed model effectively improves both prediction accuracy and perceptual quality; 2) The distance-based frame sampling strategy enhances model performance; 3) The LPIPS loss contributes to improved visual quality; and 4) End-to-end training proves to be effective.

### 4.3. End-to-End training

To avoid local optima, we implement an end-to-end training framework while strategically blocking gradient propagation from the rectified flow model to the precipitation estimator, thereby preventing the flow model from interfering with the estimator's learning of physical motion dynamics. Since the precipitation estimator consistently converges before the rectified flow model during training, it can continuously provide the rectified flow model with augmented samples from the $i$-th frame, thereby enhancing the robustness of the rectified flow model.

The training and inference processes of the model are outlined in Algorithm 1 and Algorithm 2. In the training phase, the precipitation estimator first generates the posteriori sequence $\hat{Y}^*$ and computes the MSE loss, then stops the gradient of $\hat{Y}^*$ and inputs $\hat{Y}^*$ as the initial set to the rectified flow model, which randomly selects a posteriori frame $y$ in each round according to the frame sampling strategy, and then performs scheduling as scheduled, computes the forward paths between the posteriori frame $\hat{y}^*$ and the real frame according to the acceptance step $t$, and computes the forward

*Table 1.* Statistics of the datasets used in the experiments.

| Dataset | Size | | | Seq Len | | Spatial Resolution |
|---|---|---|---|---|---|---|
| | Train | Valid | Test | In | Out | $H \times W$ |
| SEVIR | 13020 | 1000 | 2000 | 13 | 36 | $128 \times 128$ |
| MeteoNet | 8640 | 500 | 1500 | 13 | 36 | $128 \times 128$ |

*Table 2.* Quantitative evaluation of state-of-the-art spatiotemporal prediction algorithms on Sevir and MeteoNet benchmarks.

| Method | SEVIR | | | | | MeteoNet | | | | |
|---|---|---|---|---|---|---|---|---|---|---|
| | CSI | HSS | SSIM | MSE | LPIPS | CSI | HSS | SSIM | MSE | LPIPS |
| MAU | 0.241 | 0.312 | 0.705 | 0.0093 | 0.348 | 0.197 | 0.281 | 0.819 | **0.0029** | 0.250 |
| ConvLSTM | 0.240 | 0.306 | 0.711 | 0.0084 | 0.304 | 0.192 | 0.272 | 0.811 | 0.0031 | 0.256 |
| SimVP | 0.241 | 0.306 | 0.717 | 0.0083 | 0.334 | 0.165 | 0.239 | 0.774 | 0.0037 | 0.294 |
| Earthformer | 0.214 | 0.277 | 0.685 | 0.0100 | 0.374 | 0.158 | 0.224 | 0.765 | 0.0035 | 0.303 |
| Earthfarseer | 0.209 | 0.262 | 0.675 | **0.0077** | 0.385 | 0.161 | 0.232 | 0.789 | 0.0030 | 0.306 |
| STRPM | 0.213 | 0.281 | 0.621 | 0.0130 | 0.322 | 0.154 | 0.223 | 0.701 | 0.0064 | 0.322 |
| CasCast | 0.238 | 0.301 | 0.709 | 0.0120 | 0.285 | 0.183 | 0.274 | 0.810 | 0.0062 | 0.252 |
| DiffCast | 0.244 | 0.318 | 0.692 | 0.0100 | 0.286 | 0.199 | 0.287 | 0.816 | 0.0054 | 0.241 |
| PercpCast | **0.267** | **0.360** | **0.722** | 0.0092 | **0.268** | **0.209** | **0.305** | **0.820** | 0.0049 | **0.237** |

*Table 3.* Quantitative evaluation of state-of-the-art spatiotemporal prediction algorithms on Sevir and MeteoNet benchmarks with different thresholds for CSI.

| Method | SEVIR | | | | | MeteoNet | | | | |
|---|---|---|---|---|---|---|---|---|---|---|
| | CSI74 | CSI133 | CSI160 | CSI181 | CSI219 | CSI16 | CSI24 | CSI32 | CSI36 | CSI40 |
| MAU | 0.484 | 0.206 | 0.096 | 0.064 | 0.020 | 0.348 | 0.236 | 0.092 | 0.039 | 0.023 |
| ConvLSTM | **0.497** | 0.203 | 0.085 | 0.052 | 0.013 | 0.349 | 0.218 | 0.079 | 0.031 | 0.021 |
| SimVP | 0.490 | 0.177 | 0.090 | 0.064 | 0.026 | 0.307 | 0.181 | 0.053 | 0.022 | 0.011 |
| Earthformer | 0.453 | 0.165 | 0.066 | 0.040 | 0.003 | 0.308 | 0.173 | 0.031 | 0.010 | 0.002 |
| Earthfarseer | 0.469 | 0.110 | 0.048 | 0.029 | 0.012 | 0.300 | 0.176 | 0.041 | 0.018 | 0.009 |
| STRPM | 0.424 | 0.164 | 0.084 | 0.059 | 0.025 | 0.289 | 0.183 | 0.084 | 0.049 | 0.022 |
| CasCast | 0.440 | 0.193 | 0.105 | 0.067 | 0.023 | 0.315 | 0.228 | 0.108 | 0.043 | 0.020 |
| DiffCast | 0.449 | 0.203 | 0.113 | 0.084 | 0.032 | 0.323 | 0.242 | 0.127 | 0.053 | 0.024 |
| PercpCast | 0.496 | **0.251** | **0.134** | **0.099** | **0.037** | **0.354** | **0.276** | **0.132** | **0.068** | **0.027** |

### 5.1. Experimental Setting

**Datasets. SEVIR**: The SEVIR dataset (Veillette et al., 2020) includes satellite images, NEXRAD VIL radar echograms, and lightning data from 20,393 weather events. Each event spans 384 km × 384 km, with 49 images taken every 5 minutes over a 4-hour sequence. The first 13 images are used to predict the subsequent 36.

**MeteoNet**: The MeteoNet dataset (Larvor et al., 2020) consists of radar and satellite images, ground-based observations, and meteorological data. Each event covers an area of 550 km × 550 km, with 49 images captured every 5 minutes in a 4-hour sequence. As with SEVIR, the first 13 images are used to predict the next 36.

Both datasets are divided into training, validation, and test sets. The statistical data for all datasets used in the experiments are summarized in Table 1. The images are rescaled to the range [0, 1] and normalized.

**Evaluation** The accuracy metrics include CSI, HSS and

MSE. CSI quantifies the proportion of correct predictions, reflecting the model's event identification ability. HSS compares the model's predictions to random guessing, measuring its forecasting skill. MSE calculates the mean squared error between the predicted and the real value. Their formulas are as follows:

$$\text{CSI} = \frac{TP}{TP + FN + FP} \quad (15)$$

where $TP$ is the number of true positives, $FP$ is the number of false positives, and $FN$ is the number of false negatives.

$$\text{HSS} = \frac{TP \times TN - FN \times FP}{(TP+FN)(FN+TN)+(TP+FP)(FP+TN)} \quad (16)$$

where $TN$ is the number of true negatives.

$$\text{MSE} = \frac{1}{n}\sum_{i=1}^{n}(y_i - \hat{y}_i)^2 \quad (17)$$

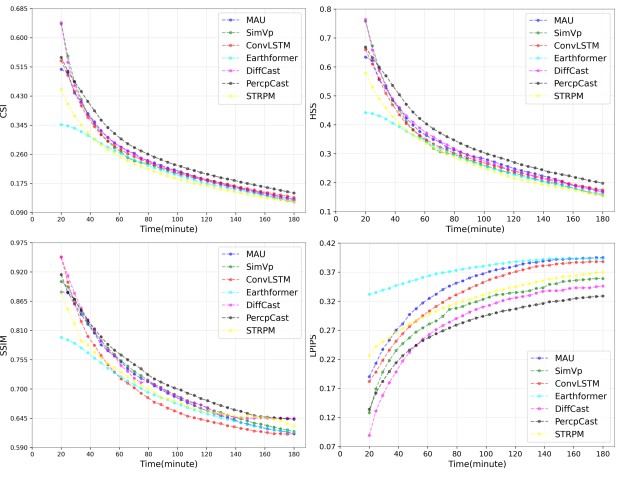

*Figure 3.* Performance changes of CSI, HSS, SSIM and LPIPS at different moments in time on SEVIR.

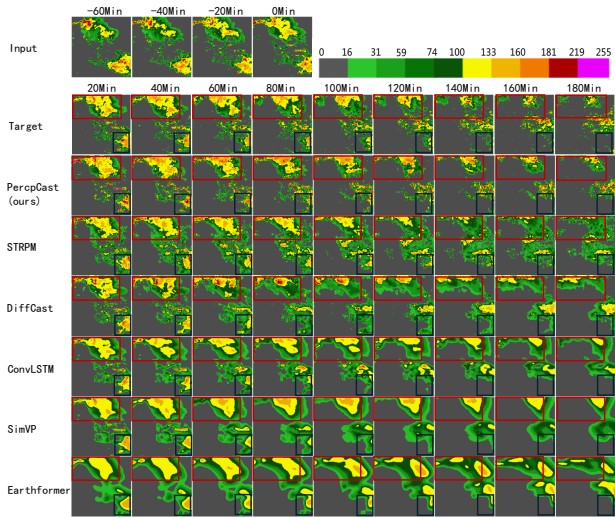

*Figure 4.* A visual comparison example on a precipitation event from SEVIR. More cases are in the Appendix D.

where $y_i$ is the true value and $\hat{y}_i$ is the predicted value.

Image perception metrics include SSIM and LPIPS. SSIM evaluates perceptual similarity based on brightness, contrast, and structure, and is related to the overall structure of the image. LPIPS uses deep learning models to evaluate perceptual similarity, focuses on human visual perception, and is related to image smoothing.

$$\text{SSIM}(x,y) = \frac{(2\mu_x\mu_y + C_1)(2\sigma_{xy} + C_2)}{(\mu_x^2 + \mu_y^2 + C_1)(\sigma_x^2 + \sigma_y^2 + C_2)} \quad (18)$$

where $\mu_x$ and $\mu_y$ are the mean values of the image patches, $\sigma_x^2$ and $\sigma_y^2$ are the variances, and $\sigma_{xy}$ is the covariance.

$$\text{LPIPS}(x,y) = \frac{1}{N}\sum_{i=1}^{N}\|f_i(x) - f_i(y)\|_2^2 \quad (19)$$

where $f_i(x)$ and $f_i(y)$ are the features extracted from the $i$-th layer of a pre-trained neural network.

**Implementation Details** We employ a cosine learning rate schedule to train the model, with a maximum learning rate of 1e-4 and a minimum learning rate of 1e-7. The warm-up ratio is set to 20%, with the warm-up learning rate set to 3e-4. The model is trained for 100 epochs using the AdamW optimizer. A detailed analysis of the model's hyperparameters is presented in the next section.

**Baselines** To validate the effectiveness of the proposed model, we selected several notable works as comparison baselines, including SimVP (Gao et al., 2022b), Earthformer (Gao et al., 2022a), MAU (Chang et al., 2021), ConvLSTM (Shi et al., 2015), DiffCast (Yu et al., 2024), CasCast(Gong

et al., 2024), STRPM (Chang et al., 2022) and Earthfarseer (Wu et al., 2024).

**5.2. Experimental Results**

From the results presented in Table 2, Table 3, Table4, and Figure 3, the following key observations can be made: (1) Compared to state-of-the-art models, our framework demonstrates marked enhancements in both prediction accuracy and perceptual quality, achieving average increases of 5% in CSI and 10% in LPIPS metrics, with pooled CSI showing 2-9% improvements across test scenarios. This result underlines the merits of the proposed approach, in particular its balanced performance in terms of accuracy and perceived quality. As for mse, our method performs optimally in the generative model (STRPM,DiffCast), less than twice mse of any method. (2) The accuracy improvement of our model is especially notable in high echo regions, where the prediction outcomes are enhanced by 15% to 22% compared to other models for CSI thresholds greater than 74 and greater than 24. This suggests that our model exhibits superior robustness in handling high echo regions. (3) As the prediction time increases, our model continues to maintain the highest perceptual quality and prediction accuracy compared to other models, demonstrating its stability and advantage in long-term forecasting.

In general, the deterministic models (SimVP, ConvLSTM) of the compared methods tend to output a sequence of posteriori mean values, which are susceptible to high echo value decay with increasing prediction time, leading to perfor-

*Table 4.* Quantitative evaluation of state-of-the-art spatiotemporal prediction algorithms on Sevir and MeteoNet benchmarks with different Pooled CSI.

| Method | SEVIR | | | Meteonet | | |
|---|---|---|---|---|---|---|
| | P1 | P4 | P16 | P1 | P4 | P16 |
| MAU | 0.241 | 0.268 | 0.285 | 0.197 | 0.231 | 0.260 |
| ConvLSTM | 0.240 | 0.266 | 0.292 | 0.192 | 0.236 | 0.264 |
| SimVP | 0.241 | 0.263 | 0.283 | 0.165 | 0.196 | 0.214 |
| Earthformer | 0.214 | 0.254 | 0.265 | 0.158 | 0.189 | 0.207 |
| Earthfarseer | 0.209 | 0.252 | 0.267 | 0.161 | 0.193 | 0.212 |
| STRPM | 0.213 | 0.236 | 0.271 | 0.154 | 0.190 | 0.203 |
| CasCast | 0.238 | 0.262 | 0.289 | 0.183 | 0.207 | 0.231 |
| DiffCast | 0.244 | 0.270 | 0.294 | 0.199 | 0.235 | 0.265 |
| PercpCast | **0.267** | **0.287** | **0.299** | **0.209** | **0.240** | **0.268** |

*Table 5.* Analysis of different training strategies on SEVIR.

| METHOD | CSI | HSS | SSIM | LPIPS |
|---|---|---|---|---|
| WITH GRADINET | 0.258 | 0.338 | 0.694 | 0.292 |
| TWO STAGE | 0.249 | 0.321 | 0.694 | 0.289 |
| LPIPS(0.0) | 0.256 | 0.328 | 0.701 | 0.324 |
| LPIPS(0.5) | 0.267 | 0.360 | 0.722 | 0.268 |
| LPIPS(1.0) | 0.265 | 0.358 | 0.711 | 0.272 |
| κ=0.00 | 0.262 | 0.348 | 0.703 | 0.278 |
| κ=0.02 | 0.263 | 0.343 | 0.709 | 0.276 |
| κ=0.05 | 0.267 | 0.360 | 0.722 | 0.268 |
| κ=0.1 | 0.266 | 0.346 | 0.705 | 0.280 |

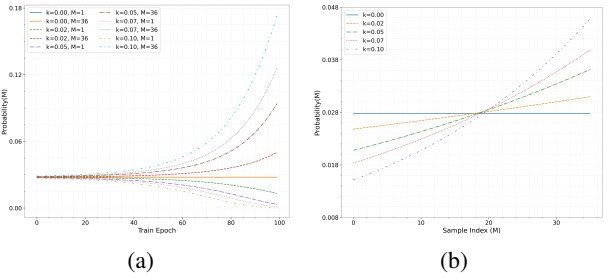

*Figure 5.* Sampling probability: (a) probability over time at different k values. (b) probability per frame in the 60th epoch at different k values

mance degradation. On the other hand, while probabilistic models account for the modelled data distribution, the STRPM model suffers from an excessive number of degrees of freedom, preventing it from effectively capturing future changes. The residual nature of DiffCast limits its focus to local distributions, while its ability to capture global information is insufficient.

In Figure 4, we present and compare the results of different methods for the precipitation forecasting task. The cases represent three typical precipitation processes: precipitation attenuation and separation (highlighted in the red box), precipitation dissipation (in the black box), and precipitation formation (between the red and black boxes). Overall, ConvLSTM suffers from cumulative errors, leading to a rapid decay in intensity (red box). SimVP captures the attenuation and dissipation processes well but struggles to model precipitation formation (between the red and black boxes). Earthformer effectively models the formation of the intermediate rainbands as a transfer of precipitation within the black box, but its prediction is too widespread in the red box. STRPM and DiffCast both preserve good precipitation details, but STRPM exhibits overly broad precipitation extent and a rapid intensity decay (120-180 min) in the red

box. DiffCast's residual nature leads it to focus mainly on local distributions, limiting its ability to capture global information and resulting in poor performance for longer-term forecasts (red box).

### 5.3. Analysis and Discussions

**Is end-to-end training preferable to two-stage training?** The present model has been trained in an end-to-end manner, aiming to optimize both the precipitation estimator and the rectified flow module simultaneously. In theory, the model could also be trained in a two-stage manner, where the optimal precipitation estimator is first determined, then frozen, followed by modeling the posteriori mean sequence using a rectified flow model. To validate the performance of these two training methods, we experimented with the two-stage training strategy. The results, shown in Table 5, reveal that our end-to-end approach outperforms the two-stage method. During end-to-end training, the gradient transfer between the precipitation estimator and the rectified flow model is stopped, ensuring that the performance of the precipitation estimator remains stable, regardless of the training approach. By treating the posteriori mean sequence as conditional, the rectified flow model can learn a mapping from the data distribution (i.e., the posteriori frame distribution) to the target frame distribution at each index. As training progresses, the precipitation estimator gradually converges, and the posteriori frame at each position in each iteration becomes a sample of the data distribution at that position, enhancing the robustness of the rectified flow model. Additionally, experiments with end-to-end training without gradient stopping also outperformed the two-stage method, despite not meeting the theoretical requirements.

**Is the loss of Lpips necessary?** We used Lpips loss to constrain the perceptual ability of the flow model at each time step during the transport process. We conducted experiments with different weights for the LPIPS loss, setting them to 0, 0.5, and 1. The experimental results are shown

in Table 5, where the perceptual quality of the model decreases by 22% and the prediction performance drops by 5% when the LPIPS loss weight is set to 0. When the LPIPS loss weights are set to 0.5 and 1, the metrics remain essentially unchanged. Additional experiments with varying loss weights can be found in Appendix C.

**Analysis of different hyperparameters of frame sampling strategies.** We employ the distance-weighted sampling strategy in an attempt to focus the rectified flow model on longer predicted frames. This strategy involves two hyperparameters: the weight growth factor $k$ and the temperature parameter $t$. The temperature parameter $t$ is empirically set with a maximum value of $t_{\max} = 50$ and a minimum value of $t_{\min} = 0.5$. We then investigate the impact of the weight growth factor $k$ on training performance. Notably, $k$ is typically sampled uniformly as it approaches 0; the greater the deviation of $k$ from 0, the higher the probability that distant samples will be selected. To evaluate this, we conducted experiments with sampling strategies for $k = 0$, $k = 0.02$, $k = 0.05$, and $k = 0.1$, respectively. The probability density curves for these values of $k$ are shown in Figure 5. As illustrated in Table 5, appropriate distance weighting improves the prediction accuracy and perceptual quality of long-distance frames. However, excessive emphasis on distant frames reduces attention to other frames with distributional differences (e.g., intermediate frames), leading to a degradation in performance. Further experiments with hyperparameters are provided in Appendix C.

## 6. Conclusion

This paper presents a precipitation nowcasting method based on perceptual constraints, addressing the challenge that traditional approaches struggle to effectively capture future uncertainty in long-term sequences. The method transforms posteriori mean sequences into a real distribution using rectified flow and introduces a frame sampling strategy to enhance focus on frames further into the future. Experimental results demonstrate that the proposed model outperforms existing techniques on publicly available radar datasets, validating its effectiveness and reliability.

## Limitations and Future work

PercpCast achieves competitive performance on different datasets, but it may not be able to predict sudden convective storms, that is, storms that suddenly form precipitation without storm signals in the initial stage. Improving such predictions requires incorporating atmospheric variables such as temperature, humidity, and wind direction during the precipitation formation process. Our future work aims to explore how to integrate atmospheric variables to build a unified prediction precipitation model.

## Acknowledgements

This work was supported in part by NSFC under Grants 62376072, 62272130, and in part by Shenzhen Science and Technology Program No. KCXFZ20240903093006009, KCXFZ20211020163403005, ZDSY20120613125016389.

## Impact Statement

The primary focus of our work is technological advancements and improving machine learning algorithms. As such, our work does not directly involve ethical considerations or has immediate societal consequences.

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

# A. Analysis and Proof

According to (Freirich et al., 2021; Blau & Michaeli, 2018), Eq. 2 can be obtained by solving the optimal transport problem,

$$p_{U,V} \in \arg \min_{p_{U',V'} \in \Pi(p_Y, p_{\hat{Y}^*})} \mathbb{E}[\|U' - V'\|^2], \quad (20)$$

i.e., given an observation $X$, first predict the posteriori mean $\hat{Y}^* = \mathbb{E}[Y|X = x]$, and then get the optimal transport plan from $p_{\hat{Y}^*}$ to $p_Y$, which get the twice minimal mean square error(2MMSE).

Similar to (Freirich et al., 2021), in precipitation forecasting, we can assume that given the historical precipitation $X$, the real precipitation $Y$ and the estimated precipitation $\hat{Y}$ are independent of each other, and $\hat{Y}^*$ is the optimal solution for the posteriori mean, and then we can get $\hat{Y} \leftarrow \hat{Y}^* \leftarrow X \rightarrow Y$, which can satisfy the above process for the two-stage training, but in the end-to-end training, the $\hat{Y}$ can affect $\hat{Y}^*$, and although this may not affect our ability to construct the optimal transfer from $\hat{Y}^*$ to $Y$, it does impact the analysis of property b. Therefore, to make the above process satisfy the independence principle, we stop the gradient backpropagation from $\hat{Y}$ to $\hat{Y}^*$.

For this optimal transport problem, (Liu et al., 2022) proposed a flow matching algorithm based on straight lines transportation. The algorithm is composed of two processes: the forward process is given by: $Z_t = tZ_1 + (1-t)Z_0$ and the backward process is: $v_\theta(Z_t, t) = \mathbb{E}[Z_1 - Z_0|Z_t]$, (Liu et al., 2022) noted that solving the ODE with $v_\theta$ provides an approximation of the optimal transport map from the source distribution to the target distribution, especially when the process is repeated multiple times (i.e., reflow). To learn $v_\theta$, the model is trained by minimizing the following loss:

$$\int_0^1 \mathbb{E}\left[\|(Y_1 - Y_0) - v_\theta(Y_t, t)\|^2\right] dt, \quad (21)$$

where $Y_1 = Y, Y_0 = \hat{Y}^*$. According to (Freirich et al., 2021) and Eq. 21, we get Eq. 3. From Eq. 3, we can get these properties as follows:

(a) $p_{\hat{Z}_1} = p_Y$

(b) **The MSE of $\hat{Z}_1$ is smaller than the MSE of the solution in Eq. 20.**

Proof. (a) We assume Eq. 22 exists a unique solution. From Theorem 3.3 in (Liu et al., 2022), we have $p_{\hat{Z}_t} = p_{Z_t}$ for every $t \in [0, 1]$. This implies that $p_{\hat{Z}_1} = p_{Z_1} = p_Y$, i.e. our method satisfies the constraints of Eq. 2.

$$d\hat{Z}_t = v_\theta(\hat{Z}_t, t) dt, \quad \text{with} \quad \hat{Z}_0 = Z_0 = \hat{Y}^*. \quad (22)$$

(b) From (Freirich et al., 2021), $E[\|Y - \hat{Y}'\|^2]$ can be decomposed as $E[\|Y - \hat{Y}^*\|^2] + E[\|\hat{Y}' - \hat{Y}^*\|^2]$. Since $p_{\hat{Y}', \hat{Y}^*} = p_{Y, \hat{Y}^*}$, it follows that $E[\|\hat{Y}' - \hat{Y}^*\|^2] = E[\|Y - \hat{Y}^*\|^2]$, it means $E[\|Y - \hat{Y}'\|^2] = 2E[\|Y - \hat{Y}^*\|^2] = 2MMSE$. Following arguments similar to those used in the proof of Theorem 3.5 in (Liu et al., 2022), it holds that

$$\mathbb{E}[\|\hat{Z}_1 - \hat{Y}^*\|^2] = \mathbb{E}\left[\left(\int_0^1 v_{RF}(\hat{Z}_t, t) \, dt\right)^2\right] \quad (23)$$

$$= \mathbb{E}\left[\left(\int_0^1 v_{RF}(Z_t, t) \, dt\right)^2\right] \quad (24)$$

$$\leq \mathbb{E}\left[\int_0^1 \|v_{RF}(Z_t, t)\|^2 \, dt\right] \quad (25)$$

$$= \mathbb{E}\left[\int_0^1 \mathbb{E}[(Y - \hat{Y}^*)^2|Z_t] \, dt\right] \quad (26)$$

$$\leq \mathbb{E}\left[\int_0^1 \mathbb{E}[\|Y - \hat{Y}^*\|^2|Z_t] \, dt\right] \quad (27)$$

$$= \int_0^1 \mathbb{E}\left[\mathbb{E}[\|Y - \hat{Y}^*\|^2|Z_t]\right] dt \quad (28)$$

$$= \int_0^1 \mathbb{E}[\|Y - \hat{Y}^*\|^2] \, dt \quad (29)$$

$$= \mathbb{E}[\|Y - \hat{Y}^*\|^2], \quad (30)$$

Equation (23) follows from the definition of $\hat{Z}_1$ and $\hat{Y}^*$. Equation (24) follows from the fact that $p_{\hat{Z}_t} = p_{Z_t}$. Equation (25) follows from Jensen's inequality. Equation (26) follows from the definition of $v_\theta(Z_t, t)$. Equation (27) follows from Jensen's inequality. Equation (28) follows from the linearity of the integral operator. Equation (29) follows from the law of total expectation, then we get $\mathbb{E}[\|\hat{Z}_1 - \hat{Y}^*\|^2] \leq \mathbb{E}[\|Y - \hat{Y}^*\|^2]$. Since $\hat{Z}_1$ is the final output under the independence assumptions, we can get the conclusion by using the following equation:

$$\mathbb{E}\left[\left\|Y - \hat{Z}_1\right\|^2\right] = \mathbb{E}\left[\left\|Y - \hat{Y}^*\right\|^2\right] + \mathbb{E}\left[\left\|\hat{Z}_1 - \hat{Y}^*\right\|^2\right]$$
$$\leq 2\mathbb{E}\left[\left\|Y - \hat{Y}^*\right\|^2\right] = 2\text{MMSE}, \quad (31)$$

which means that the MSE of our method is less than 2 MMSE.

# B. Datasets

**SEVIR**: The SEVIR (Storm EVent ImagRy) dataset, as described by Veillette(Veillette et al., 2020), includes visible and infrared satellite images, NEXRAD VIL radar echograms, and surface lightning events. This study focuses on VIL radar echograms from 20,393 weather events

recorded between 2017 and 2020. Each event spans a 384 km × 384 km area and 1 km spatial resolution, with a 4-hour sequence consisting of 49 images taken at 5-minute intervals.

For the analysis, 49 consecutive images from each event are used to predict the next 36 VIL images (180 minutes) based on the first 13 observed images (65 minutes). The data is split into training (October 2018–August 2019), validation (September–October 2019), and test (October–November 2019) sets. The images are rescaled to the range of 0-255 and binarized using specific thresholds[16,74,133,160,181,219]. Performance is evaluated using the Critical Success Index (CSI), Heidke Skill Score (HSS), Structural Similarity Index (SSIM), Learned Perceptual Image Patch Similarity (LPIPS), and Mean square error (MSE), with mean values reported.

**MeteoNet**: The MeteoNet dataset, as described by Larvor(Larvor et al., 2020), is a multimodal dataset that includes satellite and radar imagery, ground-based observations, and meteorological data. It covers a 550 km × 550 km area and 1 km spatial resolution in the northwest of France, with each 4-hour sequence consisting of 49 images taken every 5 minutes.

For the analysis, 49 consecutive images from each event are used to predict the next 36 VIL images (180 minutes) based on the first 13 observed images (65 minutes). The data is divided into training (October 2016–August 2019), validation (September–October 2019), and test (September–October 2019) sets. The images are rescaled to the range of 0-70 and binarized using specific thresholds[8,16,24,32,36,40]. Performance is evaluated using the CSI, HSS, SSIM, LPIPS, and MSE, with mean values reported.

## C. More Experiment Analysis

Table 6. Results of different weights of the losses on SEVIR.

| $(\mathcal{L}_{pe}, \mathcal{L}_{rf}, \mathcal{L}_{lpips})$ | CSI | HSS | SSIM | LPIPS | MSE |
|---|---|---|---|---|---|
| (0, 1, 0.5) | 0.044 | 0.312 | 0.311 | 0.369 | 0.0217 |
| (1, 0, 0.5) | 0.240 | 0.307 | 0.663 | 0.233 | 0.0085 |
| (1, 1, 0.0) | 0.256 | 0.328 | 0.701 | 0.324 | 0.0102 |
| (2, 1, 0.5) | 0.266 | 0.360 | 0.717 | 0.269 | 0.0091 |
| (1, 2, 0.5) | 0.264 | 0.355 | 0.712 | 0.270 | 0.0093 |
| (1, 1, 0.5) | 0.267 | 0.360 | 0.722 | 0.268 | 0.0092 |
| (1, 1, 1.0) | 0.265 | 0.358 | 0.711 | 0.272 | 0.0094 |

**Weighted of Losses**

There are there losses in our method, $\mathcal{L}_{pe}$, $\mathcal{L}_{rf}$ and $\mathcal{L}_{lpips}$, in Table 6 and Figure 6, We analysed the weights of the three losses. In our experiments, firstly, because the gradients of the previous and subsequent parts of our model are

separated, the weights of $\mathcal{L}_{pe}$ and $\mathcal{L}_{rf}$ do not have much effect on the training, secondly, we set the weight of $\mathcal{L}_{pe}$ to 0 and do not use supervised signals for the precipitation estimator, which results in the training not being able to converge, and then, we set the weight of $\mathcal{L}_{lpips}$ to 0, which can lead to a tessellated artefacts problem, and lastly, the rectified flow part will be not be used, the Although lower lpips are obtained the lack of effective estimation of the data distribution leads to poor mse and accuracy. In fact, in precipitation prediction, image quality assessment is a complex task, simply using LPIPS does not accurately reflect the structural information of the image, but lpips is the most important factor in it, who is directly related to whether the image is smooth or not, which is the main problem solved by our model. Since diversity metrics such as FID are not applicable to this problem, in order to get a better image quality assessment we suggest combining metrics such as SSIM at the same time, on the sevir dataset, our experience suggests that a relatively good visual quality can be obtained with LPIPS below 0.29 and SSIM above 0.7.

**Scale factor K**

Table 7. Results of different K on SEVIR.

| K | CSI | HSS | SSIM | LPIPS |
|---|---|---|---|---|
| 0.00 | 0.262 | 0.348 | 0.703 | 0.278 |
| 0.02 | 0.263 | 0.343 | 0.709 | 0.276 |
| 0.05 | 0.267 | 0.360 | 0.722 | 0.268 |
| 0.07 | 0.266 | 0.352 | 0.716 | 0.265 |
| 0.1 | 0.266 | 0.346 | 0.705 | 0.280 |
| 0.2 | 0.250 | 0.327 | 0.682 | 0.292 |

In our experiments k represents the level of focus on differently located frames, and we analysed the magnitude of different. Table 7, Figure 7 and Figure 9 shows that overfocusing on distant frames leads to performance degradation.

Table 8. Results of different scheduler on SEVIR.

| (SCHEDULER) | CSI | HSS | SSIM | LPIPS | MSE |
|---|---|---|---|---|---|
| LINEAR | 0.262 | 0.354 | 0.701 | 0.269 | 0.0089 |
| EXPONENTIAL | 0.267 | 0.360 | 0.722 | 0.268 | 0.0092 |

**Temperature** We experimented with both linear and exponential temperature scheduling, and the results of the experiments are shown in Table 8 and Figure 8

**Rectflow** We performed 1-rectflow according to (Liu et al.,

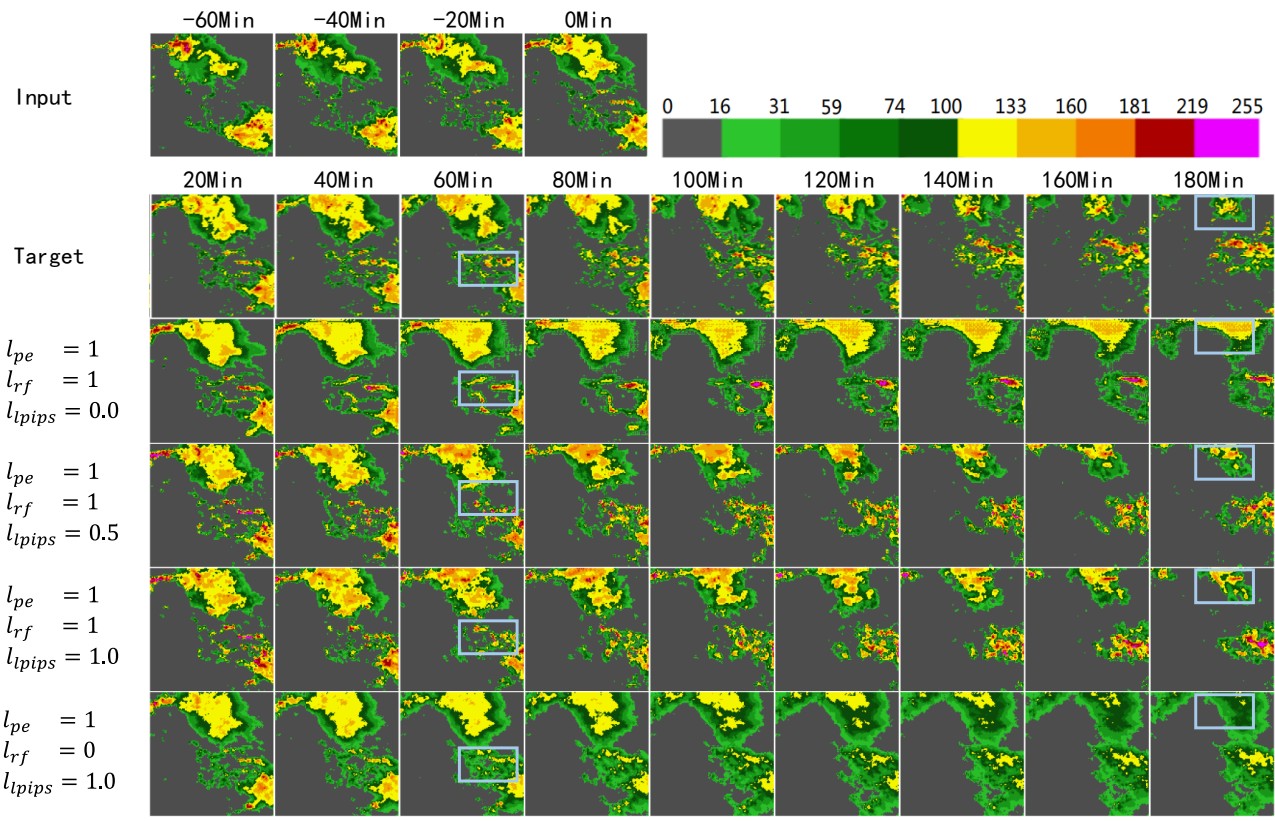

*Figure 6.* Cases of different loss weights for a precipitation event from SEVIR.

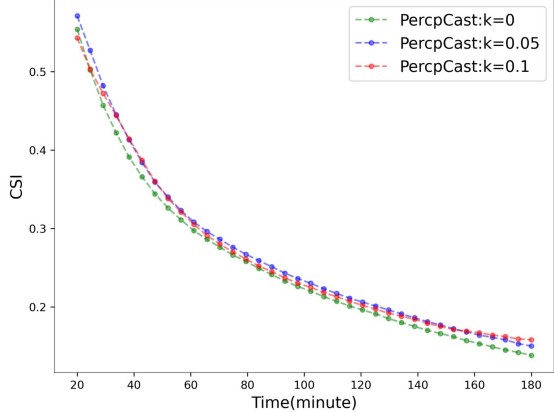

*Figure 7.* Performance changes of CSI at different moments in time on SEVIR.

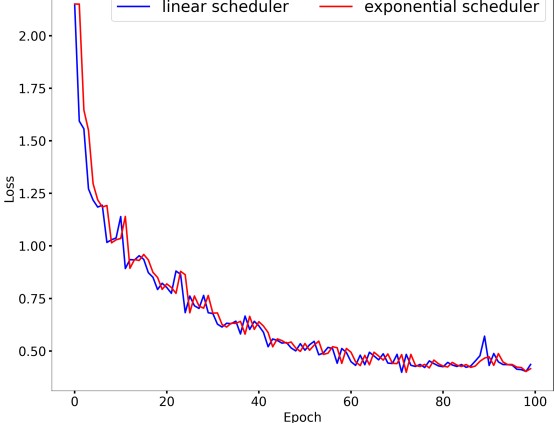

*Figure 8.* scheduler loss on SEVIR

2022) and the results of the experiment are shown in Table 9 and Figure 10.

**About the Precipitation Estimator**

For the choice of the precipitation estimator, we considered two aspects: firstly, stability, which ensures a continuous frame to frame transition and can be consistent with the physical motion, and secondly, ease of training, we evaluated a variety of models as the precipitation estimator, and found that some of the models are unstable in training and rely heavily on the selection of hyperparameters, overall, we chose the ConvLSTM as the precipitation estimator. in

*Table 9.* Results of 1-rectified flow on SEVIR.

| REFLOW | CSI | HSS | SSIM | LPIPS | MSE |
|---|---|---|---|---|---|
| 1-RECTIFIED FLOW | 0.263 | 0.356 | 0.717 | 0.265 | 0.0089 |
| WITHOUT RECTIFIED FLOW | 0.267 | 0.360 | 0.722 | 0.268 | 0.0092 |

Table 10 and Figure 11, We give the forecast performance and a case of SimVP as a precipitation estimator.

*Table 10.* Results of the precipitation estimation with SimVP on SEVIR.

| PRECIPITATION ESTIMATOR | CSI | HSS | SSIM | LPIPS | MSE |
|---|---|---|---|---|---|
| SIMVP | 0.246 | 0.321 | 0.701 | 0.297 | 0.0091 |
| CONVLSTM | 0.267 | 0.360 | 0.722 | 0.268 | 0.0092 |

# D. More Precipitation Case

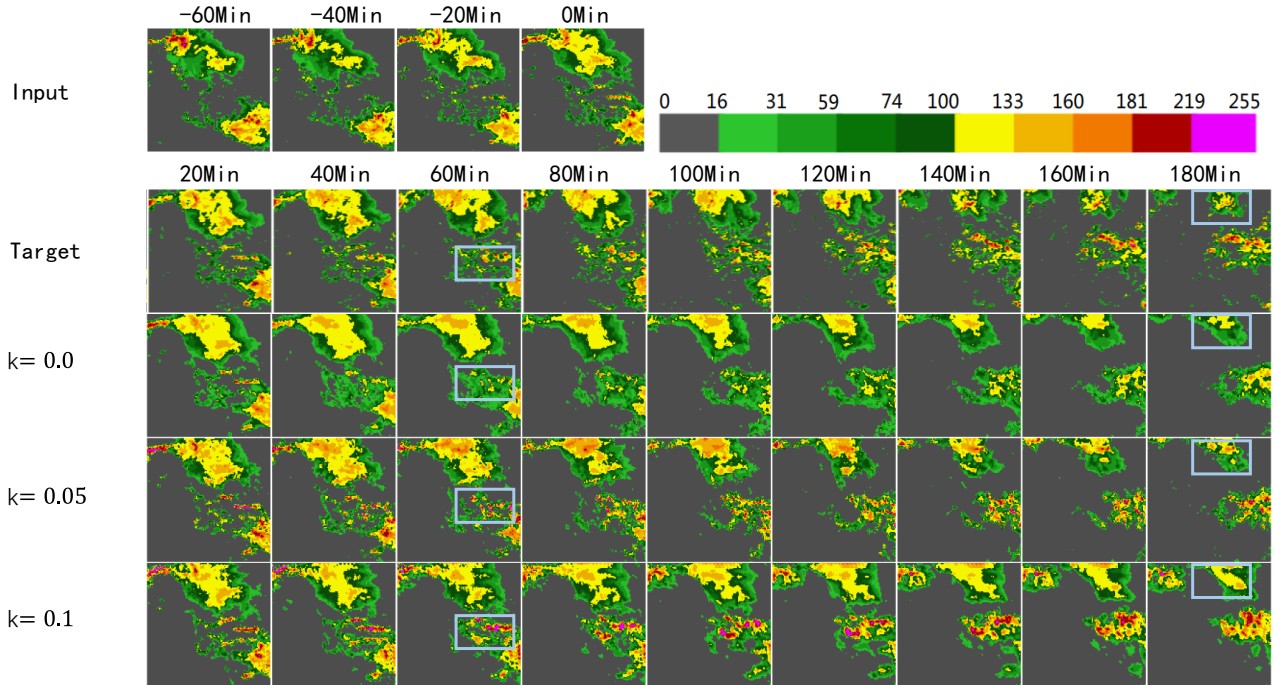

*Figure 9.* Cases of different k for a precipitation event from SEVIR.

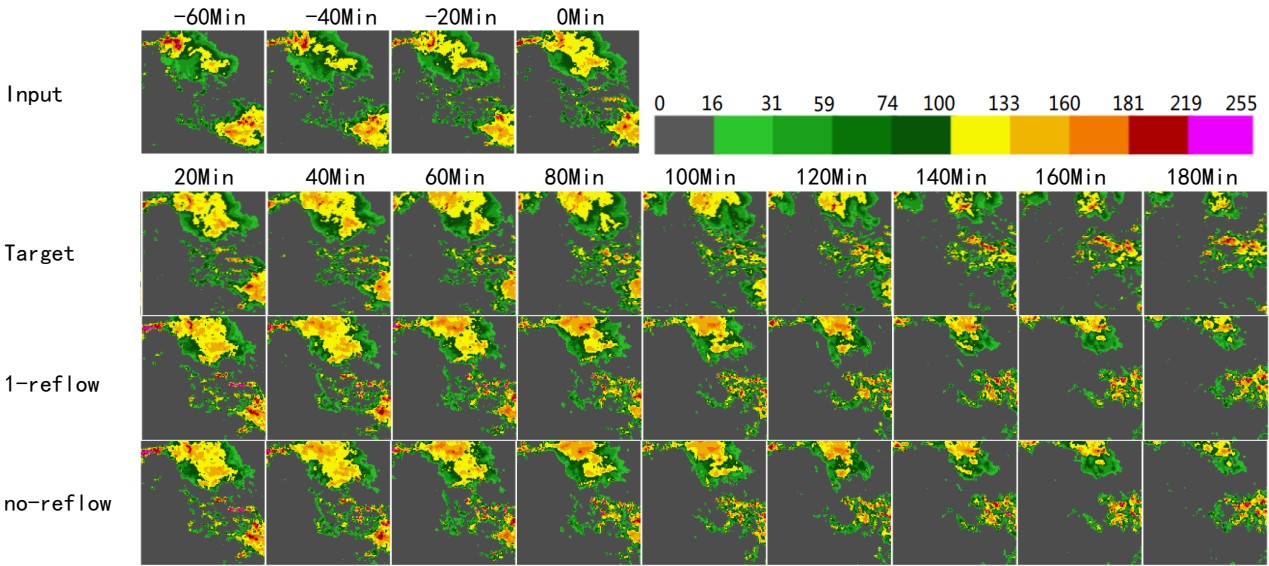

*Figure 10.* Cases of 1-reflow for a precipitation event from SEVIR.

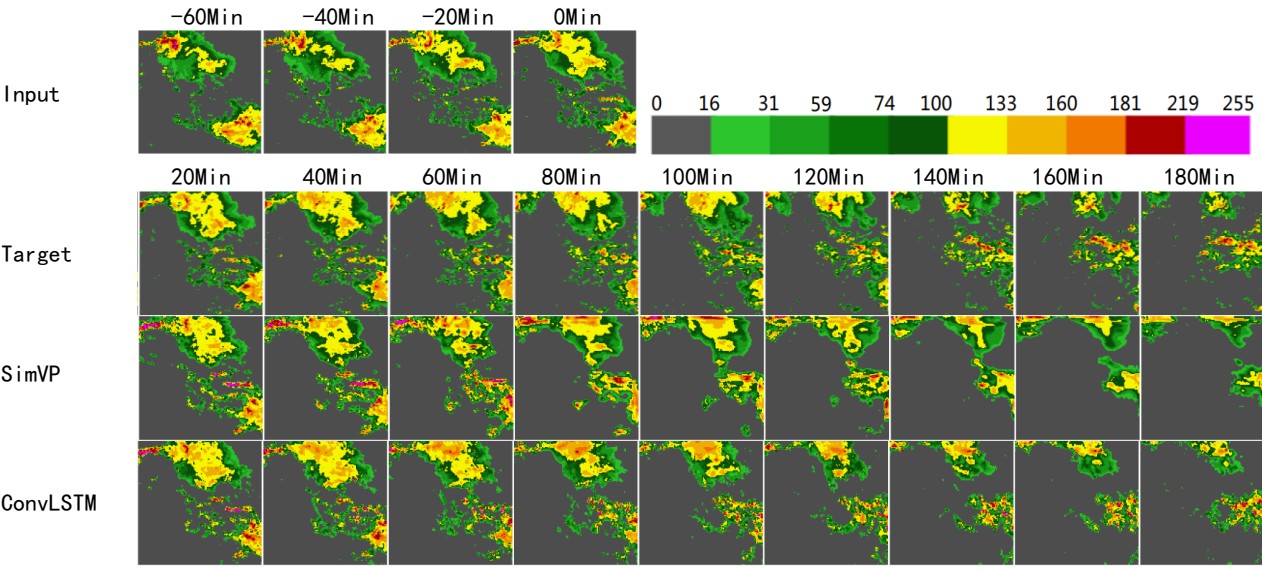

*Figure 11.* Cases of precipitation estimation with SimVP for a precipitation event from SEVIR.

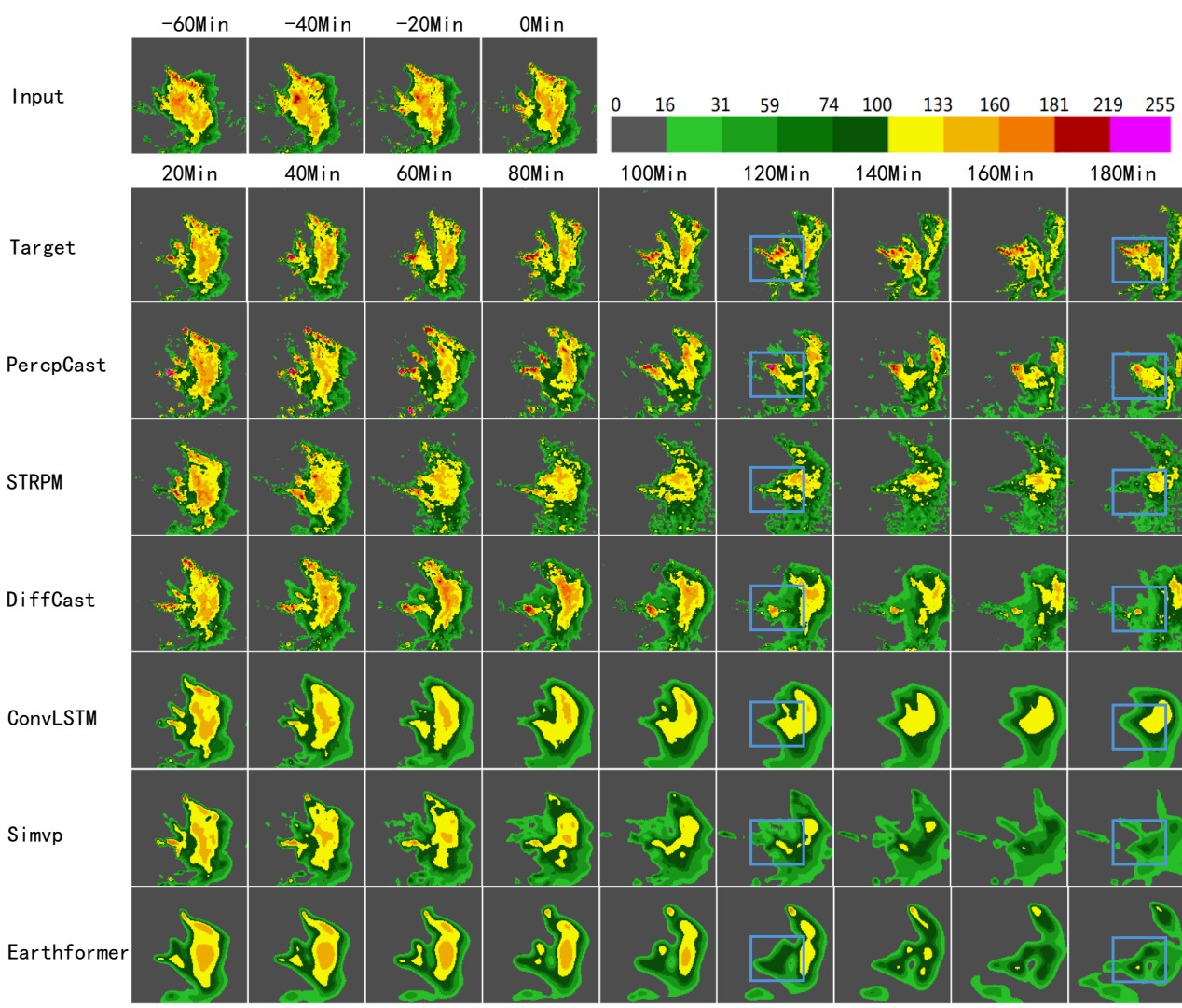

*Figure 12.* Cases of different methods for a precipitation event from MeteoNet.

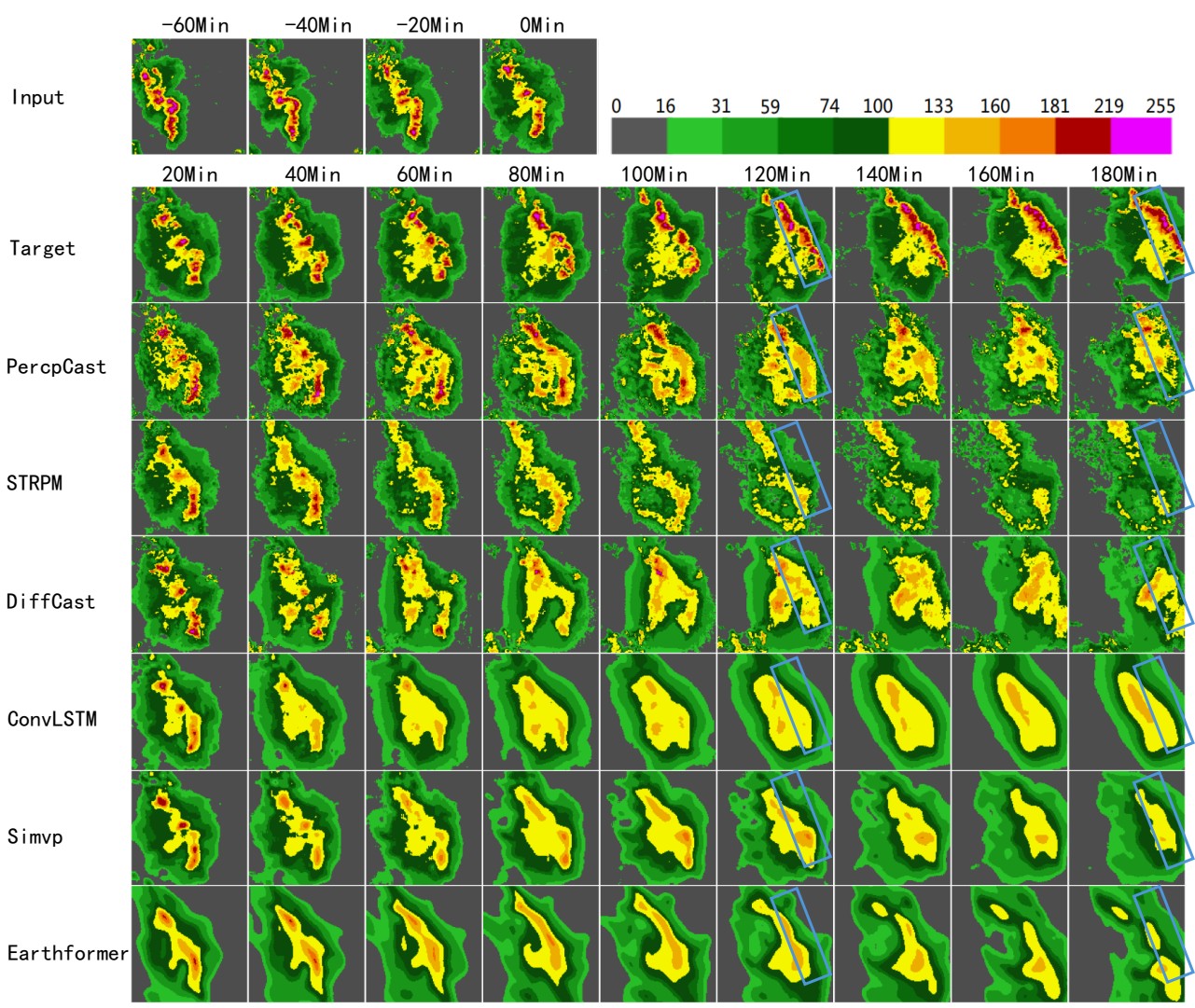

*Figure 13.* Cases of different methods for a precipitation event from SEVIR.

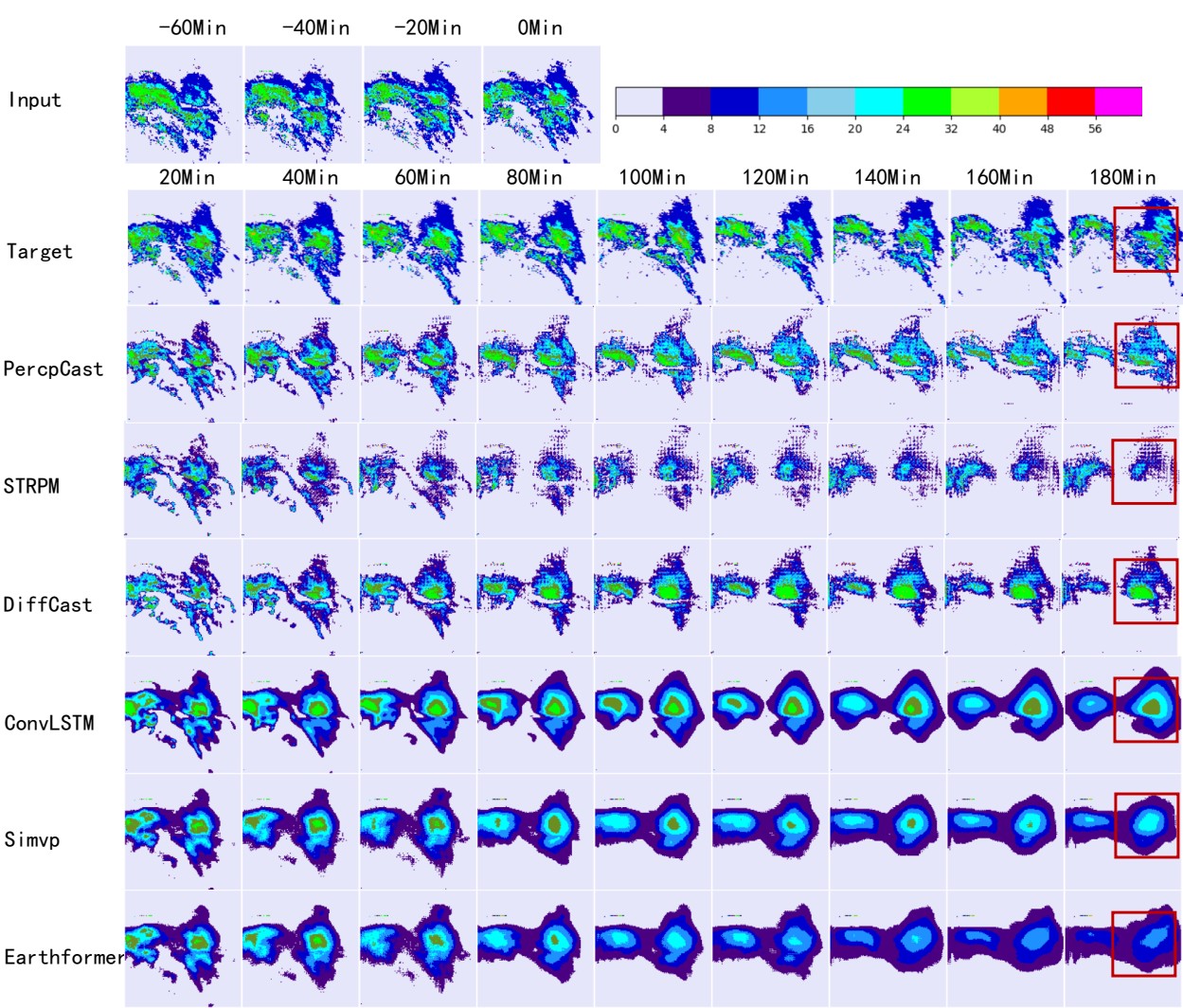

*Figure 14.* Cases of different methods for a precipitation event from MeteoNet.

