# OpenReview forum: "Perceptually Constrained Precipitation Nowcasting Model"
_ICML.cc/2025/Conference — ICML 2025 poster_

### Official Review · Reviewer_XBn6 · 2025-03-09

**Overall Recommendation:** 3

**Summary:**

This paper proposes a model called PercpCast for precipitation nowcasting, aiming to predict future rainfall patterns more accurately while also improving how realistic those predictions appear.  The authors use a two-stage approach in a single end-to-end framework: first, they generate a "posteriori mean" sequence of future precipitation using a ConvLSTM-based estimator, then refine those estimates through a "rectified flow" module that better aligns the predicted distributions with the ground truth. To address the challenge of longer-horizon forecasting, the authors introduce a frame-sampling strategy that assigns more weight to frames further in the future. The model incorporates an LPIPS-based loss function to enforce perceptual consistency. Experiments on radar datasets (SEVIR and MeteoNet) show that proposed method outperforms various existing approaches in terms of accuracy metrics and visual quality metrics.

**Claims And Evidence:**

The paper's main claims about improved accuracy and perceptual quality are largely supported by experiments on two types of datasets.
However, some points could benefit from clearer evidence or explanation. Specifically, it is not entirely clear why the authors stop the gradient transfer between the precipitation estimator and the rectified flow model. Additionally, it is unclear how the authors decided on the specific weight for the LPIPS loss.

**Essential References Not Discussed:**

No additional foundational works appear to be missing.

**Experimental Designs Or Analyses:**

The experiments use known radar datasets and standard precipitation metrics, aligning with typical nowcasting  research. The train/validation/test splits are standard, and the comparisons with multiple baselines are appropriate. However, ablation studies on the LPIPS loss weight or frame sampling would further validate the design choices.

**Methods And Evaluation Criteria:**

The authors use recognized datasets (SEVIR, MeteoNet) and metrics (CSI, HSS, MSE, SSIM, LPIPS) that directly relate to precipitation forecasting and capture both accuracy and perceptual realism.

**Other Comments Or Suggestions:**

Suggestions are addressed in the “Questions for Authors” section

**Other Strengths And Weaknesses:**

Strength: PercpCast integrates a precipitation estimator with a rectified flow model to achieve forecasts that are both accurate and visually realistic. The model's frame-sampling strategy, which emphasizes distant frames, makes it particularly effective for long-horizon predictions. The proposed work is supported by thorough evaluations on established public datasets and comparisons with multiple baselines.

Additionally, the paper combines theoretical explanations with visual demonstrations, providing a well-rounded view of its strengths.
Weakness:

One notable weakness is the lack of clarity regarding why gradient transfer is stopped between the precipitation estimator and the rectified flow model. The paper does not clearly explain how this decision aligns with the assumption that the predicted and true frames are independent, which leaves an important theoretical justification underexplored.

Additionally, the paper omits details about the hardware specifications used during training, making it challenging to assess the method's computational requirements compared to simpler baselines.

**Questions For Authors:**

1. The author often labels the ConvLSTM’s output as a "posteriori mean sequence", but does not clearly explain why it represents the average of future rainfall. A brief note on how minimizing MSE leads the model to predict an "average" outcome would make this point clearer.

2. In Section 5.3, the authors states "During end-to-end training, the gradient transfer between the precipitation estimator and the rectified flow model is  stopped....". Could you clarify why this is done and how it aligns with the assumption that the predicted and true frames are independent, given the historical data?

3. Please include details about the hardware specifications on which the model was trained. It would help readers compare the resource requirements of the proposed method with simpler baselines.

4. In Figure 4, PercpCast appears to overestimate precipitation in certain areas (indicated by the pink colors), compared to the ground-truth maps. Could you clarify what might cause these overestimations.

5. The paper mentions using LPIPS loss to make the predictions look more realistic, but it's not clear why a specific weight was chosen for it in the loss function. Did the authors run any experiments to figure out the best value, or was it selected based on intuition?

6. The frame sampling strategy gives different importance to frames depending on how far they are in time. Could the authors share any experimental results or tests that show how this choice affects the model's accuracy, especially for longer predictions?

**Relation To Broader Scientific Literature:**

The paper extends ongoing research in precipitation nowcasting, which traditionally separates into deterministic approaches that focus on reducing mean-squared error, and probabilistic approaches that aim for more realistic detail. By combining a precipitation estimator with a rectified flow model, this work bridges both views: it maintains the long-term accuracy of deterministic models while incorporating the realistic detail of generative methods. The introduction of a frame-sampling strategy also connects to broader ideas in temporal modeling tasks.

**Theoretical Claims:**

In appendix, the authors includes a theoretical derivation that connects the precipitation nowcasting objective to an optimal transport framework. The derivation appears logically consistent with prior work on rectified flows.

---

> ### Author Rebuttal · Authors · 2025-04-01
>
> We appreciate the reviewer’s detailed feedback. We will address their concerns and eager to engage in a more detailed discussion with the reviewer.
>
>
> ### **Q1.**
>
> Thank you for the comment. In precipitation nowcasting, the high uncertainty in short-term evolution means a single historical observation can correspond to multiple future scenarios. ConvLSTM minimizes the mean squared error loss$\mathcal{L}_{pe}=\mathbb{E}\left[\|Y-\hat{Y}^*\|^2\right]$, causing predictions  $\hat{Y}^*$  to converge to the conditional expectation $\mathbb{E}[Y|X]$. This produces a posterior mean sequence – a statistical average of all possible future precipitation outcomes under given input conditions.
>
> ### **Q2 & W1**
>
> In Section 3 and Appendix A, we outlined the gradient-stopping operation. To clarify: Due to diverging learning objectives between the precipitation estimator and the rectified flow model, gradient stopping is applied to prevent the rectified flow model from interfering with the estimator's acquisition of physical motion dynamics. This constraint forces the rectified flow model to actively learn physical consistency (e.g., motion continuity and distribution alignment) directly from input data. Meanwhile, end-to-end training improves model robustness and alleviates suboptimal solutions inherent in two-stage frameworks.
>
> Regarding independence assumptions, our approach aligns with Freirich et al. (2021) in theoretical framework but diverges in the training methodology. Under the condition of stopping gradient propagation from $\hat{Y}^*$ to $\hat{Y}$, the generation of $\hat{Y}$ does not influence $\hat{Y}^*$. This establishes a Markov process: $\hat{Y} \leftarrow \hat{Y}^* \leftarrow X \rightarrow Y$. Consequently, given historical data $X$, $Y$ remains independent of both $\hat{Y}$ and $\hat{Y}^*$, while $\hat{Y}$ depends solely on $\hat{Y}^*$. This ensures compliance with the predictive independence assumption with theoretical justification, achieving causal decoupling between variables.
>
> ### **Q3 & W2.**
>
> Thank you for the comment. Our model employs mixed-precision training (FP16) on a single NVIDIA A100 80GB GPU, with supporting hardware including an Intel(R) Xeon(R) Platinum 8350C CPU @ 2.60GHz. Resource utilization metrics of our method compared to the ConvLSTM on the SEVIR dataset are summarized in the following Table.
>
>
> | Method              | Parameters (M) | GPU Memory (Batch Size=1,GB) | Training Time(hour) |  Inference Time (Batch Size=1,Second) |
> |---------------------|---------------:|----------------------:|------------------------:|------------------------:|
> | Proposed Model      |          55.87 |                   4.1 |                     16  |                       2.47  |
> | ConvLSTM            |          17.81 |                   3.4 |                     11  |                     1.92  |
>
>
> ### **Q4.**
>
> To understand the issue, we can see Figure 9 that short-term predictions are relatively easier due to closer alignment between predicted and real frame distributions, while long-term predictions suffer significant distribution drift - a discrepancy amplified by frame sampling strategies that prioritize learning long-term variations, thereby compromising short-term prediction accuracy. As shown in the Figure 9,  the issue can be alleviated by using a moderate small $k$. However, a smaller $k$ may reduce long-term prediction accuracy. Hence, it is a trade-off.
>
>
> ### **Q5.**
>
> We present the experimental results of different LPIPS loss weight configurations in Table 4, Table 5, and Figure 6. These demonstrate that incorporating the LPIPS loss effectively suppresses checkerboard artifacts in the rectified flow model. Notably, within a reasonable range(0.5~1), the specific weighting configurations do not significantly affect the experimental outcomes. Detailed results for additional weight configurations will be supplemented as the table below.
>
> | lpips weight | CSI   | HSS   | SSIM  | LPIPS | MSE    |
> |------|------:|------:|------:|------:|-------:|
> | 0.0         | 0.256 | 0.328 | 0.701 | 0.324 | 0.0102 |
> | 0.2         | 0.260 | 0.342 | 0.714 | 0.287 | 0.0098 |
> | 0.5         | 0.267 | 0.360 | 0.722 | 0.268 | 0.0092 |
> | 0.7         | 0.265 | 0.357 | 0.718 | 0.265 | 0.0095 |
> | 1.0         | 0.265 | 0.358 | 0.711 | 0.272 | 0.0094 |
> ---
>
>
>
>
> ### **Q6.**
> The experimental results and visualisations (Tables 4 and 6; Figures 5 and 7-9) show our exponential sampling strategy, where K determines the range of variation of the sampling probability. Figure 5 shows the k-probability relationship, while Table 4 evaluates different k settings. Figure 7 confirms larger k improves distant frame prediction accuracy. Figure 8 compares linear and exponential sampling. Experiments show that with more iterations, moderate increases in distant frame sampling probability improve long-term prediction. However, over-amplification undermines learning of other frames, causing performance drops.

---

### Official Review · Reviewer_xKo8 · 2025-03-13

**Overall Recommendation:** 4

**Summary:**

This work proposes PercpCast, integrating both Precipitation Estimator (Video prediction model) and the Rectified Flow module. Rectified Flow module learns the transmission from the distribution of the posterior mean predicted by Precipitation Estimator to the distribution of ground truth. Further, LPIPS regularization is introduced in addition to the two typical loss terms for the Precipitation Estimator and Rectified Flow Modules. Besides, temperature-distance weighted scheduling is implemented to ensure the model focuses on the later frames. With all these techniques, PercpCast showcases its effectiveness in the outlined evaluation setting across two radar echoes datasets: SEVIR and MeteoNet.

**Claims And Evidence:**

Most claims are supported by the quantitative result in Table 2 and 3.

**Essential References Not Discussed:**

Most essential references are discussed. It will be better to compare a few more diffusion-based models with PercpCast, such as PreDiff and CasCast (Gong et al., ICML 2024) since they have a closer structure with PercpCast.

**Experimental Designs Or Analyses:**

The experimental design is generally fine, except a minor problem:

- LPIPS is chosen to be one of the evaluation metrics. Meanwhile, it is introduced as a regularization term during the training of PercpCast. This might have fairness issues compared with other baselines. The authors can consider using FVD or pooled CSI as an alternative to LPIPS like the PreDiff paper.

**Methods And Evaluation Criteria:**

The proposed methods and evaluation criteria makes sense for this problem. To verify the generalisation of PercpCast, its performance is evaluated across two datasets and compared with several SOTA.

**Other Comments Or Suggestions:**

- Some information is not very consistent. In Table 1, the output sequence length is shown to be 49, but in the main text it is described to be 36. Does that mean the PercpCast model also reconstructs the input beside forecasting the future?
- The writing in the appendix is quite messy, especially in Appendix C. Please proofread and fix.
- A lot of previous works (PreDiff, DiffCast, CasCast, etc.) also report a pooled CSI with different thresholds to evaluate the “skillfulness” of the forecasts. Observing the tables in the papers, realistic and clear forecasts tend to have higher pooled CSI. This will also be a good indicator to replace LPIPS.

**Other Strengths And Weaknesses:**

This section summarizes the strengths and weaknesses discussed above.

**Strengths:**
- Using Rectified Flow to learn the distribution difference between the posterior mean and the ground truth is quite a new idea in this task.
- It showcases its remarkable performance compared with SOTA in the perspectives of perceptuality and accuracy.

**Weaknesses:**
- The current evaluation scheme (LPIPS) might be unfair.
- A few minor but confusing parts in the appendix.

Overall, this paper delivers an interesting solution to precipitation nowcasting. Judging from the good performance results, I am inclined to accept the paper.

---
### Update after rebuttal
The authors mostly addressed my concerns and adopted my suggestions. I will keep the recommendation.

**Questions For Authors:**

Questions are asked in the above sections.

**Relation To Broader Scientific Literature:**

This work adopts a common strategy of using a precipitation estimator and fine-tuning module for precipitation nowcasting like DiffCast and CasCast. Different from previous works like DiffCast and PreDiff which utilize diffusion models in the second stage, the use of the Rectified Flow Module presents a similar but novel idea to model the difference in distribution. I believe this will very much benefit future studies.

**Theoretical Claims:**

The use of Rectify Flow is well grounded by ample previous works. There is no concern on the attempt here. However, I am quite confused on what do author attempt to show in Appendix A by proving $\mathbb{E}[ || \hat{Z}_1 - \hat{Y}^{\*} ||^2] \leq \mathbb{E}[ || Y - \hat{Y}^{*} ||^2]$. The said MSE is between the final prediction and **precipitation estimation’s output**, not the ground truth. Do the authors intend to use this to show that equation 20 is small enough so that equation 2 can also be satisfied?

---

> ### Author Rebuttal · Authors · 2025-04-01
>
> We are grateful for the reviewer's acknowledgment of our work ​and their detailed feedback, which will help us refine our research.
>
> ### **Theoretical Claims.**
> Equation (2) can be solved through either Equation (20) or our proposed method, which has different error bounds. Freirich et al. established and showed that the theoretical optimal solution for Equation (20) is 2MMSE. Here we show the error bound of our method is smaller than 2MMSE by proving $\\mathbb{E}\\left[\\left\\|\hat{Z}_1-\\hat{Y}^*\\right\\|^2\right] \\leq \\mathbb{E}\\left[\\left\\|Y-\\hat{Y}^*\\right\|^2\\right]$ in Appendix A. Since $\\hat{Z}_1$ is the final output ​under the independence assumptions, we can get the conclusion by using the following equation:
>
> $\\begin{aligned} \\mathbb{E}\\left[\\left\\|Y-\\hat{Z}_1\\right\\|^2\\right] & =\\mathbb{E}\\left[\\left\\|Y-\\hat{Y}^*\\right\\|^2\\right]+\\mathbb{E}\\left[\\left\|\\hat{Z}_1-\\hat{Y}^*\\right\|^2\\right] \\ & \\leq 2 \\mathbb{E}\\left[\\left\\|Y-\\hat{Y}^*\\right\\|^2\\right]=2MMSE\\end{aligned}$
>
> ### **W1 & S3.**
>
> Thank you for the advice. We incorporate additional experiments with CasCast and reported the pooled CSI prediction results as shown in the following table. The experimental results further validate the effectiveness of our method. The results will be updated in the revised manuscript.
>
>
> | Method         | SEVIR              |                     |                     | Meteonet            |                     |                     |
> |----------------|-------------------:|-------------------:|-------------------:|-------------------:|-------------------:|-------------------:|
> |                | Pool1              | Pool4              | Pool16             | Pool1              | Pool4              | Pool16             |
> | MAU            |              0.241 |              0.268 |              0.285 |              0.197 |              0.231 |              0.260 |
> | ConvLSTM       |              0.240 |              0.266 |              0.292 |              0.192 |              0.236 |              0.264 |
> | SimVP          |              0.241 |              0.263 |              0.283 |              0.165 |              0.196 |              0.214 |
> | Earthformer    |              0.214 |              0.254 |              0.265 |              0.158 |              0.189 |              0.207 |
> | Earthfarseer   |              0.209 |              0.252 |              0.267 |              0.161 |              0.193 |              0.212 |
> | STRPM          |              0.213 |              0.236 |              0.271 |              0.154 |              0.190 |              0.203 |
> | CasCast        |              0.238 |              0.262 |              0.289 |              0.183 |              0.207 |              0.231 |
> | DiffCast       |              0.244 |              0.270 |              0.294 |              0.199 |              0.235 |              0.265 |
> | PercpCast      |          ​**0.267** |          ​**0.287** |          ​**0.299** |          ​**0.209** |          ​**0.240** |          ​**0.268** |
>
>
>
>
> ### **W2 & S2.**
>
> Thanks for your careful review. We have reviewed Appendix C and corrected the experimental results in Tables 5–6, which will be updated as follows:
>
>
>
> | (Lpe, Lrf, Llpips) | CSI   | HSS   | SSIM  | LPIPS | MSE    |
> |---------------------|------:|------:|------:|------:|-------:|
> | (0, 1, 0.5)         | 0.044 | 0.312 | 0.311 | 0.369 | 0.0217 |
> | (1, 0, 0.5)         | 0.240 | 0.307 | 0.663 | 0.233 | 0.0085 |
> | (1, 1, 0.0)         | 0.256 | 0.328 | 0.701 | 0.324 | 0.0102 |
> | (2, 1, 0.5)         | 0.266 | 0.360 | 0.717 | 0.269 | 0.0091 |
> | (1, 2, 0.5)         | 0.264 | 0.355 | 0.712 | 0.270 | 0.0093 |
> | (1, 1, 0.5)         | 0.267 | 0.360 | 0.722 | 0.268 | 0.0092 |
> | (1, 1, 1.0)         | 0.265 | 0.358 | 0.711 | 0.272 | 0.0094 |
>
>
>
> | $K$  | CSI   | HSS   | SSIM  | LPIPS  |
> |-------|------:|------:|------:|-------:|
> | 0.00  | 0.262 | 0.348 | 0.703 | 0.278  |
> | 0.02  | 0.263 | 0.343 | 0.709 | 0.276  |
> | 0.05  | 0.267 | 0.360 | 0.722 | 0.268  |
> | 0.07  | 0.266 | 0.352 | 0.716 | 0.265  |
> | 0.1   | 0.266 | 0.346 | 0.705 | 0.280  |
> | 0.2   | 0.250 | 0.327 | 0.682 | 0.292  |
> ### **S1.**
>
> Thank you for identifying this issue. The precipitation estimator ​reconstructs the input 13 frames and ​predicts 36 future frames, resulting in a total sequence length of 49. To eliminate ambiguity, we will revise Table 1 as follows:
>
>
> | Dataset   |       |Size |      |    Seq Len|    | Spatial Resolution |
> |-----------|----------|--------|--------|-------|-----------|---------|
> |           | Train | Valid | Test | In | Out | H × W      |
> | SEVIR     | 13020 |  1000 | 2000 | 13 |  36 | 128 × 128 |
> | MeteoNet  |  8640 |   500 | 1500 | 13 |  36 | 128 × 128 |

---

### Official Review · Reviewer_PwA2 · 2025-03-14

**Overall Recommendation:** 3

**Summary:**

This article proposes a new precipitation forecasting model PercpCast, which introduces perceptual constraints into precipitation forecasting tasks. This method first uses ConvLSTM as a precipitation estimator to obtain the posterior mean sequence of future frames. Then, a module based on "rectified flow" is used to adjust the distribution of the posterior mean sequence to the distribution of the real target frame. Finally, a distance weighted frame sampling strategy is used to further enhance the attention to future frames. The experimental part was thoroughly validated on two public datasets, SEVIR and MeteoNet, and the results showed that the method exhibited certain advantages in perceptual quality (LPIPS, SSIM) and event detection metrics (CSI, HSS) while maintaining a low mean square error (MSE).

**Claims And Evidence:**

Yes, the problem being solved is the inaccurate landing point of gan and diffusion in precipitation, that is, the inability to balance CSI and image quality index LIPIS.

**Essential References Not Discussed:**

No.

**Experimental Designs Or Analyses:**

1. In the innovation of the paper, it is mentioned that the current refined Gan and Diffusion have random sampling, which cannot balance CSI and image quality LIPIS (poor CSI, good LIPIS). Under the perceptual constraints of LIPIS, the second stage of flow matching can move towards a determined path towards the target distribution, reducing the accuracy of high echo landing points. The obvious difference between this method and Diffusion is that in the second stage, flow matching is used instead of Diffusion to refine the model, lacking the ability to ablate Diffusion and your Rectified Flow Model when using CnovLSTM as a precipitation estimator. This makes it difficult to verify the advantages of flow matching in balancing CSI and LPIPS, and it is unclear whether it is temperature weight weighting, lip loss, or flow matching performance that leads to the advantages of balancing CSI and LPIPS.
2. The comparison method in quantitative indicators uses MAU and Earthfarseer, There is no visual comparison between these two models in the visualization (Figure 3 and Figure 4, as well as the visualization in the supplementary materials)

**Methods And Evaluation Criteria:**

Yes, it does.

**Other Comments Or Suggestions:**

Fig. 4 with 'preparation study (in the blue box)' appears to be a black box, not a blue box.

**Other Strengths And Weaknesses:**

Strength:
The innovation lies in the use of stream matching in the second stage, which assigns greater weights to frames with longer lead times as the forecast progresses, resulting in better forecast performance after 1 hour compared to other models. The motivation is clear.

Weakness:
Some comparison methods are relatively old and lack comparison with some updated typical SOTA methods.

**Questions For Authors:**

See the above questions.

**Relation To Broader Scientific Literature:**

The problem being solved is the inaccurate landing point of gan and diffusion in precipitation, that is, the inability to balance CSI and image quality index LIPIS.

**Theoretical Claims:**

The images are rescaled to the range [0, 1] and binarized. "Are you sure about binarized? Because the forecast is based on values in the range of 0-255, Binarized doesn't look right.

---

> ### Author Rebuttal · Authors · 2025-04-01
>
> Thanks for the reviewer's valuable suggestions. We will try to address the reviewer's concerns and are eager to engage in a more detailed discussion with the reviewer.
> ###  **Theoretical Claims**.
> Thank you for pointing out this issue. We perform ​normalization (not binarization) to rescale images to [0, 1]: SEVIR (0–255) is divided by 255, and MeteoNet (0–70) by 70. We will correct the 'binarized' with 'normalized' in the revised manuscript.
> ### **Experimental Designs Or Analyses 1**
> We would like to clarify that unlike diffusion models reconstructing precipitation predictions via conditional integration, our method employs end-to-end learning to directly optimize the posterior mean sequence distribution from the precipitation estimator. To enhance this framework, we introduce two key components: ​temperature-weighted scaling and ​LPIPS perceptual loss. Comprehensive ablation studies demonstrate: (1) LPIPS regularization successfully suppresses checkerboard artifacts in rectified flow, enhancing visual coherence(Tables 4 & 5); (2) Temperature weighting significantly improves long-term frame prediction accuracy(Tables 4 & 6); (3) The rectified flow module achieves exceptional modeling of data distributions, generating meteorologically plausible precipitation patterns that effectively address issues such as high echo attenuation and missing details(Tables 5, Figure 4 & 6).
>
> To further compare the performance of diffusion models and Rectified Flow in precipitation prediction, we conducted experiments by replacing the Rectified Flow module with a diffusion model. Specifically, due to the instability caused by adapting end-to-end training to diffusion models, we first constructed a pre-trained precipitation estimator. While keeping other configurations unchanged, we then utilized noise and predicted frame as inputs for diffusion modeling during the frame sampling process. Additionally, we employed CasCast as a baseline comparison. CasCast is a non-end-to-end precipitation prediction framework where its first stage originally uses a Vision Transformer (ViT) for precipitation estimation, followed by a second stage that applies diffusion models for distribution refinement. In our implementation, we replaced CasCast's ViT-based precipitation estimator with a ConvLSTM model. The experimental results, presented in the following Table, further validate the effectiveness of the rectified flow model. The results will be supplemented in the revised manuscript.
>
> | Method    |       |       | SEVIR |       |       |       |       | MeteoNet |      |        |
> |-----------|------:|------:|------:|------:|------:|------:|------:|---------:|-----:|-------:|
> |                  | CSI   | HSS   | SSIM  | MSE   | LPIPS | CSI   | HSS   | SSIM     | MSE  | LPIPS  |
> | with Diffusion        | 0.223 | 0.288 | 0.697 | 0.0135| 0.297 | 0.177 | 0.269 | 0.797    | 0.0065| 0.268  |
> | CasCast          | 0.238 | 0.301 | 0.709 | 0.0120| 0.285 | 0.183 | 0.274 | 0.810    | 0.0062| 0.252  |
> | Proposed Model   | 0.267 | 0.360 | 0.722 | 0.0092| 0.268 | 0.209 | 0.305 | 0.820    | 0.0049| 0.237  |
>
> | Method    |       |       | SEVIR |       |       |       |       | MeteoNet |      |        |
> |-----------|------:|------:|------:|------:|------:|------:|------:|---------:|-----:|-------:|
> |                  |CSI74| CSI133| CSI160| CSI181| CSI219| CSI16| CSI24| CSI32| CSI36| CSI40|
> | with Diffusion   |0.437 |0.185| 0.075| 0.054 |0.021|0.299| 0.215| 0.098 |0.035| 0.022|
> | CasCast          |0.440 |0.193| 0.105| 0.067 |0.023|0.315| 0.228| 0.108 |0.043| 0.020|
> | Proposed Model   | 0.496| 0.251| 0.134| 0.099| 0.037| 0.354| 0.276| 0.132| 0.068| 0.027
>
> ### **Experimental Designs Or Analyses 2**
>
> These materials will be supplemented in the revised manuscript.
>
> ###  **W**
>
> Our experiments have included SOTA methods DiffCast (CVPR 2024), Earthfarseer (AAAI 2024). Following your suggestion, we also compare our method with CasCast (ICML 2024) as shown in above tables and the results will be updated in the revised manuscript. This ensures necessary comparisons with 2024 conference benchmarks.
>
> [1] Yu, D., Li, X., Ye, Y., Zhang, B., Luo, C., Dai, K., Wang, R.,and Chen, X. Diffcast: A unified framework via residual diffusion for precipitation nowcasting. In The IEEE/CVF Conference on Computer Vision and Pattern Recognition, 2024
>
> [2] Wu, H., Liang, Y., Xiong, W., Zhou, Z., Huang, W., Wang,S., and Wang, K. Earthfarsser: Versatile spatio-temporal dynamical systems modeling in one model. In Proceed-ings of the AAAI Conference on Artificial Intelligence, volume 38, pp. 15906–15914, 2024.
>
> [3] Gong, J., Bai, L., Ye, P., Xu, W., Liu, N., Dai, J., Yang,X., and Ouyang, W. Cascast: Skillful high-resolutionprecipitation nowcasting via cascaded modelling. In International Conference on Machine Learning, pp. 15809–15822. PMLR, 2024
> ### **S**
>
>  Thank you for identifying this inconsistency. We will correct the description from "blue box" to "black box" in the revised manuscript.

---

### Official Review · Reviewer_s3wb · 2025-03-30

**Overall Recommendation:** 4

**Summary:**

This paper proposes a precipitation forecast model based on perceptual constraints. Its main contributions include: proposing a new perspective on the precipitation forecast problem, that is, converting it into a posterior mean square error problem under specific constraints; designing a model architecture based on precipitation estimator and correction flow to predict precipitation series while maintaining its authenticity and continuity; and proposing a weighted sampling strategy for long-distance frames to improve the model's prediction ability for long-term series. Experimental results show that the model has better prediction accuracy than the existing optimal model.

**Claims And Evidence:**

The precipitation forecast model proposed in this paper is based on a new perspective and its effectiveness is demonstrated through experiments.

**Essential References Not Discussed:**

N/A

**Experimental Designs Or Analyses:**

From the paper, it is evident that the authors have considered multiple factors in their experimental design and analysis, conducting detailed comparisons and evaluations. They selected several representative baseline models for comparison and tested the model performance under different parameter settings. Additionally, the authors provided a thorough explanation of the hyperparameter selection process and presented concrete experimental results, including specific data and figures. Therefore, I believe the experimental design and analysis in this paper are sound.

**Methods And Evaluation Criteria:**

The method and evaluation criteria proposed in this paper are meaningful for solving the current precipitation forecast problem. This paper proposes a new perspective to solve the problems of existing methods in predicting long series, and adopts appropriate evaluation indicators to measure the accuracy and perceived quality of the model.

**Other Comments Or Suggestions:**

N/A

**Other Strengths And Weaknesses:**

Strengths:

A new perceptually constrained precipitation prediction model is proposed, which can effectively improve the prediction accuracy and image quality.

The residual flow structure and sparse sampling strategy are used to enhance the ability to focus on distant frames and capture future changes.

Experimental verification is carried out on two public datasets, and better performance and stability are achieved than existing methods.

Weaknesses:

The prediction effect in some extreme cases has not been analyzed in detail and needs further discussion.

The experimental results do not provide detailed parameter settings and hyperparameter adjustment processes, making it difficult to reproduce the experimental results.

**Questions For Authors:**

Please refer to the weaknesses.

**Relation To Broader Scientific Literature:**

The main contribution of this paper is the proposal of a perceptually constrained precipitation prediction model, which improves prediction accuracy and image quality by introducing perceptual constraints. This is different from the current precipitation prediction methods that only focus on minimizing the mean square error (MSE). This model addresses the limitations of existing methods by reconstructing the precipitation prediction problem and using perceptual constraints. The model also uses a sparse sampling strategy based on the attention mechanism and a residual flow structure to enhance the ability to focus on distant frames and capture future changes. These methods have better performance and stability than existing precipitation prediction methods. Therefore, the research results of this paper are meaningful for improving related research in the field of precipitation prediction.

**Theoretical Claims:**

This paper argues that the introduction of perceptual constraints can improve the performance of the current precipitation forecast model. Specifically, the model transforms the current precipitation forecast problem into a posterior mean square error problem and implements perceptual constraints by constructing a transmission between distributions. The experimental results of the model show that its performance is better than the current state-of-the-art model.

---

> ### Author Rebuttal · Authors · 2025-04-01
>
> We thank the reviewer for recognizing our ideas and theory.
>
> ###  **W1**
> Thank you for your question. Due to the introduction of perceptual constraint, our model has the advantage of accurately preserving high-value part in prediction image, which indicates extreme weather storms. As shown in Figures 4 and 13, our model accurately predicts the evolution of the heavy precipitation band (above 160) and gives reliable intensity estimates.In spite of the advantage, our model may also fail to predict sudden convective storms that develop precipitation abruptly where no storm signals appear at the beginning. Improving such predictions requires incorporating atmospheric variables including temperature, humidity, and wind patterns during precipitation formation, which is a key objective for our subsequent research. We will add necessary discussions in our final version.
> ###  **W2**
> We have elaborated on the impact of weight configurations for loss functions and distance sampling (Tables 4-6) in both the experiments and appendices. For other hyperparameters (e.g., learning rates), we identified appropriate values within the range of 1e-3 to 1e-5 and documented them in the main text(Section 5.1 Implementation Details). All hyperparameters in the model have been thoroughly specified, and the experimental code will be made publicly available on a community platform shortly as well.

---

### Decision · Program_Chairs · 2025-05-01

**Decision:**

Accept (poster)

**Comment:**

All reviewers support the acceptance of the paper, with two Accepts and two Weak Accepts. The main contribution is an end-to-end precipitation prediction model based on perceptual constraints using a RectifiedFlow framework. I agree with the reviewers that the paper has merits and therefore recommend acceptance. However, the authors are encouraged to incorporate the points discussed in the rebuttal into the final version.